

# OMCAT: Omni Context Aware Transformer

## Abstract

Large Language Models (LLMs) have made significant strides in text generation and comprehension, with recent advancements extending into multimodal LLMs that integrate visual and audio inputs. However, these models continue to struggle with fine-grained, cross-modal temporal understanding, particularly when correlating events across audio and video streams. We address these challenges with two key contributions: a new dataset and model, called OCTAV and OMCAT respectively. OCTAV (**O**mni **C**ontext and **T**emporal **A**udio **V**ideo) is a novel dataset designed to capture event transitions across audio and video. Second, OMCAT (**O**mni **C**ontext **A**ware **T**ransformer) is a powerful model that leverages RoTE (Rotary Time Embeddings), an innovative extension of RoPE, to enhance temporal grounding and computational efficiency in time-anchored tasks. Through a robust three-stage training pipeline—feature alignment, instruction tuning, and OCTAV-specific training—OMCAT excels in cross-modal temporal understanding. Our model demonstrates state-of-the-art performance on Audio-Visual Question Answering (AVQA) tasks and the OCTAV benchmark, showcasing significant gains in temporal reasoning and cross-modal alignment, as validated through comprehensive experiments and ablation studies. Our dataset and code will be made publicly available. The link to our demo page is https://om-cat.github.io.

## 1 Introduction

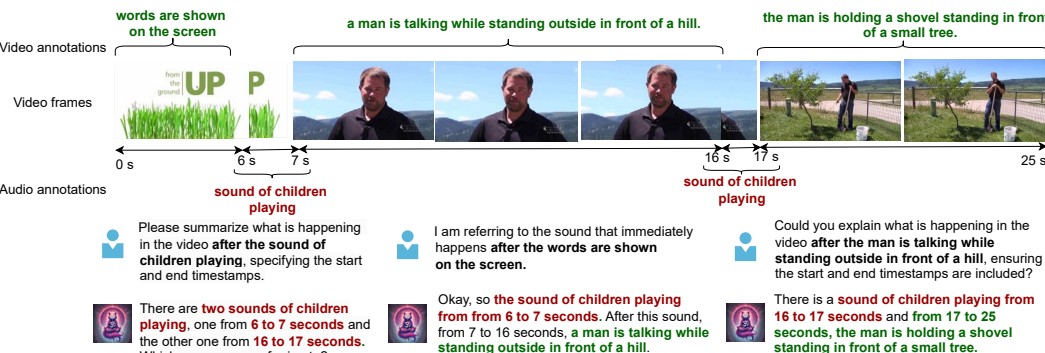

Figure 1: Illustration of a video sequence from our proposed OCTAV dataset. The annotations highlight key moments, including the timing of the audio and visual events.

Large language models (LLMs) (Achiam et al., 2023; Touvron et al., 2023) have achieved remarkable breakthroughs in both text generation and comprehension (McKeown, 1992; Achiam et al., 2023) tasks. Since then, significant progress has been made to extend LLMs to multimodal LLMs (Cheng et al., 2024; Li et al., 2023b; Maaz et al., 2023; Li et al., 2024), which integrate visual and audio inputs with textual instructions to provide understanding in multimodal contexts (Yang et al., 2022b; Chen et al., 2023a;b). These models, however, are limited in their cross-modal understanding and in their ability to provide answers to questions with fine-grained timestamps or anchored on events, as shown in Figure 1. In this paper, we address these limitations by proposing a new dataset OCTAV and

a model called `OMCAT`. The Omni Context and Temporal Audio Video dataset, `OCTAV`, consists of question-answer pairs for a video. Each question captures the transition between the events happening in the video through a sound event (*e.g.* Figure 1). The Omni Context Aware Transformer, `OMCAT`, addresses the limitations of existing models (Maaz et al., 2023; Tang et al., 2024; Su et al., 2023; Cheng et al., 2024) through a unified audio and visual language model by effectively incorporating time representations to ground the modalities temporally.

Despite the notable progress in multimodal LLMs (Li et al., 2023b; Maaz et al., 2023; Cheng et al., 2024; Lyu et al., 2023), most advancements have been centered around developing domain specific models in isolation, typically Video LLMs (Wang et al., 2023; Fu et al., 2024) or Audio LLMs (Gong et al., 2023; Kong et al., 2024; Chu et al., 2023). However, these models still face challenges in handling fine-grained, cross-modal temporal understanding when both audio and video are provided. For instance, if a user asks the question, "Is it raining in the video?" This question can be answered by either just looking at the video or listening to the audio. However, as shown in Figure 1, if the user asks the question, "Describe what happens in the video after the sound of children playing?", the model must understand both modalities because the sound of `children playing` cannot be seen, only heard, and `what the man is doing` cannot be heard, only seen. Achieving this is challenging due to several reasons, including the lack of temporally aligned cross-modal datasets, unified models and benchmarks, and clear understanding of how to combine modalities effectively.

Our goal is to achieve this cross-modal temporal understanding, and to this end we propose an instruction tuning dataset called `OCTAV`: **O**mni **C**ontext and **T**emporal **A**udio **V**ideo. Figure 1 shows a sample from our proposed `OCTAV` dataset. Existing audio and video understanding datasets (Chen et al., 2023b;a; 2020; Geng et al., 2023) only focus on open-ended question answering tasks (Yang et al., 2022b; Li et al., 2022) for audio-visual events. They lack the ability to temporally ground events or describe events that involve ambiguity or missing information in one of the modalities. Specifically, we create question-answer pairs for a video such that each question captures the transition between the events happening in the video through a sound event. For instance, as shown in Figure 1, we add the sound event of `children playing` to the silent input video between 6 to 7 seconds, during which nothing substantial happens in the video. Then, we capture the video event `before 6 seconds` and `after 7 seconds` while using the `sound of children playing` as a transition event. This setting encourages the model to not only understand the relationship between the audio and the video, but also a strong temporal understanding of both the audio and video domains in a single setup. Despite this artificial setup, our experiments show that a model trained with this data performs well in naturally occurring video and audio pairs.

While dataset design is necessary, it is not a sufficient condition to achieve cross-modal understanding given the challenges in modelling such data. As such, we propose a new approach that embeds absolute and relative temporal information in the audio and visual features, improving the model's ability to become temporally-aware. With the goal of improving cross-modal and temporal understanding, and following common practice in multimodal LLMs (Li et al., 2023b; Cheng et al., 2024; Li et al., 2024; Tang et al., 2024; Fu et al., 2024), we divide model training into 3 stages. The first training stage is focused on feature alignment, and uses audio-text, video-text, and audio-video-text data (Liu et al., 2024; Mei et al., 2024; Chen et al., 2023b). In the second stage, the model is instruction-tuned with data (Luo et al., 2023; Li et al., 2023b; Drossos et al., 2020; Chen et al., 2020) that promotes temporal and cross-modal understanding. Finally, the model is trained to support complex and cross-modal temporal data in the `OCTAV` dataset as shown in Figure 1. We name the model trained with our proposed `OCTAV` dataset and the temporal conditioning strategy `OMCAT`, for **OM**ni **C**ontext **A**ware **T**ransformer. Through this learning strategy, our method outperforms existing models on AVQA tasks (Yang et al., 2022b; Li et al., 2022) and beats baselines by a significant margin on our proposed `OCTAV` benchmark dataset.

In summary, our main contributions are as follows:
- We introduce a novel method for generating synthetic instruction-tuning dataset, `OCTAV`, which has temporal and contextual audio and video question/answer pairs addressing the limitations of existing datasets. This dataset has both training and evaluation samples to promote research in this direction.
- We propose `OMCAT`: a unified, temporally-aware audio and visual language model with fine-grained and cross-modal understanding, achieved through a staged training strategy that leverages all combinations of audio, video and text data.
- We propose `RoTE`: a simple yet efficient modification to RoPE that provides better scores on benchmarks and better computational efficiency than existing approaches for temporal conditioning,

especially on time-anchored tasks.

- Finally, we exhaustively evaluate `OMCAT`, including ablations, on a variety of multimodal tasks. Our experiments demonstrate that our model raises the standards on AVQA tasks, temporal understanding tasks and our proposed `OCTAV` benchmark.

## 2 RELATED WORK

**Multimodal LLMs.** Since the rise of large language models (LLMs) (Achiam et al., 2023; Chiang et al., 2023; Touvron et al., 2023), there has been growing interest in integrating additional modalities (Cheng et al., 2024; Gong et al., 2023; Kong et al., 2024). Video LLMs (Li et al., 2023b; Fu et al., 2024; Wang et al., 2023) utilize video-text datasets to address tasks like video question answering (Xu et al., 2016; Yu et al., 2019), visual grounding (Kazemzadeh et al., 2014), and understanding temporal segments (Gao et al., 2017; Huang et al., 2024). These have evolved into multimodal LLMs (Cheng et al., 2024; Maaz et al., 2023; Lyu et al., 2023), which encode multiple modalities and focus on coarse-grained tasks like audio-video understanding and question answering (Shu et al., 2023; Chen et al., 2023a; Yang et al., 2022b). However, these models struggle with fine-grained audio-visual tasks, where precise synchronization is key to deeper event comprehension.

Recent efforts have attempted to address this. GroundingGPT (Li et al., 2024) predicts fine-grained timestamps but is limited to sound events, while AVicuna (Tang et al., 2024) takes a more balanced approach to audio-visual temporal understanding. However, both models fall short in capturing intricate cross-modal temporal dynamics. Our work aims to address these gaps by focusing on fine-grained cross-modal information integration.

**Instruction tuning datasets.** GPT-based methods have been widely used to create datasets for video, audio, and audio-visual tasks, advancing multimodal models with large-scale resources. In video understanding, they generate and annotate datasets for tasks like video captioning (Fu et al., 2024), video question answering (Xu et al., 2016; Yu et al., 2019), and action recognition (Yu et al., 2019). Similarly, for audio understanding, instruction tuning datasets (Kong et al., 2024; Goel et al., 2024) target sound events (Salamon et al., 2014), audio captioning (Kim et al., 2019), and audio question answering (Lipping et al., 2022). Recently, AI-generated datasets have also progressed in audio-visual tasks like captioning (Chen et al., 2023a), question answering (Yang et al., 2022b), and dialog (Alamri et al., 2019). Despite this progress, current datasets remain predominantly coarse-grained, lacking fine-grained temporal and cross-modal synchronization. Our proposed dataset, `OCTAV`, addresses this limitation, enabling more precise alignment between audio and visual cues in complex scenarios.

## 3 THE `OCTAV` DATASET

One of the challenges in developing models that can understand strongly timestamped and anchored events is the lack of datasets that have this information (Wang et al., 2023; Liu et al., 2024; Chen et al., 2020; Li et al., 2023b; Tang et al., 2024; Lyu et al., 2023). To overcome this limitation, we propose a pipeline to generate a synthetic dataset called `OCTAV`, for **O**mni **C**ontext **T**emporal **A**udio **V**ideo dataset. Figure 1 shows an example from our proposed `OCTAV` dataset. First, we discuss how we identify relevant event transitions in videos. Then, we discuss how we anchor those transitions on audio samples and finally, we show how to generate question-answer pairs for these synthetically curated videos.

**Identifying transitions between video events.** To achieve this, we utilize videos with strongly timestamped captions (Zhou et al., 2018; Krishna et al., 2017; Tang et al., 2019; Zala et al., 2023), *i.e.* a video $V$ with time-caption pairs $\{(t_1, c_1), (t_2, c_2) \ldots (t_k, c_k)\}$, where $k$ is the total number of time chunks annotated in the video. Given a list of timestamped video captions indexed by $i$ and bounded by start time ($t_i^s$) and end time ($t_i^e$) each, we find pairs where the gap between end time and start time is smallest than $m$ and the sum of their lengths, from earliest to latest, is at most $T$ seconds. Empirically we set $m = 10$ and $T = 30$, ensuring that the videos are not too far apart and their length is not too long. Next, we discuss how to anchor sound between these video event transitions.

**Anchoring chunked videos on a single sound event.** For these chunked videos, we inject a sound event between the timestamp $t_i^e$ and $t_j^s$. More specifically, we randomly sample a sound event $s$ from a variety of different sound sources (Salamon et al., 2014; Fonseca et al., 2021; Piczak, 2015; Rashid et al., 2023). Details of these sound sources are provided in Appendix C.4. We remove the original audio in the given video chunk and insert this sound event between the timestamp $\{t_i^e, t_j^s\}$ to create a

strongly timed video chunk anchored on a sound event. We refer to this subset of the dataset as the `OCTAV-ST` dataset where, ST is for single-turn.

**Anchoring chunked videos on multiple sound events.** We extend the videos from a single sound event to two sound events as shown in Figure 1. Particularly, we first create a chunked video with three unique events $c_i$, $c_j$, and $c_k$ corresponding to timestamps $t_i$, $t_j$ and $t_k$ respectively, following the same procedure discussed previously. Then, we add a random sound event after removing the original audio between the timestamps $\{t_i^e, t_j^s\}$ and $\{t_j^e, t_k^s\}$. We refer to this subset with interwoven and timestamped videos with audio events as the `OCTAV-MT` dataset where, MT stands for multi-turn.

**Creating question-answer pairs.** Here, we discuss how to create question-answer pairs for the interwoven videos in the `OCTAV-ST` and `OCTAV-MT` dataset. Essentially, we have two (or three) video caption events for each chunked video and an associated audio event/sound between the video events. The model has to generate questions such that it can capture *what event is happening in the video* {*before the sound event, after the sound event*}, and *clarify which of the sound events the user is referring to while answering the question*. We use GPT-assisted (Achiam et al., 2023) generation to generate a diverse set of question-answer pairs. The prompts used are given in Appendix C.1 and Appendix C.2 and the list of instructions are given in the Appendix C.3.

Table 1: Statistics with number of videos and question-answer pairs for the `OCTAV-ST` dataset.

| OCTAV-ST | Train #Videos (QA Pairs) | Test #Videos(QA Pairs) |
|---|---|---|
| Youcook2 (Zhou et al., 2018) | 6832 | 2414 |
| ActivityNet (Krishna et al., 2017) | 16072 | 6228 |
| QueryD (Oncescu et al., 2021) | 16985 | - |
| COIN (Tang et al., 2019) | 31938 | - |
| HiREST (Zala et al., 2023) | 2408 | - |
| Total | 127,507 | 8642 |

Table 2: Statistics with number of videos and question-answer pairs for the `OCTAV-MT` dataset.

| OCTAV-MT | Train #Videos, #QA Pairs | Test # Videos, #QA Pairs |
|---|---|---|
| Youcook2 (Zhou et al., 2018) | 4296, 34330 | 1476, 11806 |
| ActivityNet Krishna et al. (2017) | 6463, 51670 | 1362, 10858 |
| UnAV-100-MT | 14698, 94916 | 2043, 9694 |
| Total | 25,457, 180,916 | 4,881, 32,358 |

**Dataset Statistics.** We utilize timestamped videos from Youcook2 (Zhou et al., 2018), QueryD (Oncescu et al., 2021), ActivityNet (Krishna et al., 2017), COIN (Tang et al., 2019), UnAV-100 (Geng et al., 2023) and, HiREST (Zala et al., 2023) datasets to create chunked videos. Essentially, we use these datasets as they have segmented annotations available for videos in diverse domains such as cooking, daily activities, scenes and instructional videos.

Overall, the `OCTAV-ST` dataset has 127,507 unique videos with single question-answer pairs for each video for training. For evaluation, we provide 2414 unique videos with question-answer pairs from the test subset of Youcook2 (Zhou et al., 2018), denoted as `OCTAV-ST`-Youcook2 and 6228 unique videos with question-answer pairs from the test subset of the ActivityNet dataset (Krishna et al., 2017), called as `OCTAV-ST`-ActivityNet. In Table 1, we show the breakdown of our proposed `OCTAV-ST` dataset in detail.

The `OCTAV-MT` dataset has 25,457 unique videos/multi-turn dialogues with a total of 180,916 single question-answer pairs for training. In Table 2, we show the detailed statistics of our proposed `OCTAV-MT` dataset. Specifically, we curate synthetic chunked videos for Youcook2 and ActivityNet and use the original videos from UnAV-100 dataset (Geng et al., 2023). The UnAV-100 dataset has timestamped audio-visual annotations from videos with real-time audio events and we convert this into question-answer pairs called the `OCTAV-MT` dataset (*e.g.* shown in Figure 7). We train and evaluate on this dataset to show `OMCAT`'s performance on in-the-wild and naturally occurring audio-visual settings. For evaluation on this multi-turn setup, we provide a total of 4818 unique videos with 32,358 question-answer pairs. Example annotations from both the `OCTAV-ST` and `OCTAV-MT` are given in Appendix D.

Table 3: Comparison of our proposed `OCTAV` dataset with other datasets with respect to modalities (audio/video), caption availability, multi-turn setup and timestamp information.

| Dataset | Audio | Video | Detailed captions | Multi-turn | Timestamps |
|---|---|---|---|---|---|
| InternVid (Wang et al., 2023) | ✗ | ✓ | ✓ | ✓ | ✓ |
| VALOR (Chen et al., 2023a) | ✓ | ✓ | ✓ | ✗ | ✗ |
| VAST (Chen et al., 2023b) | ✓ | ✓ | ✓ | ✗ | ✗ |
| VGG-Sound (Chen et al., 2020) | ✓ | ✓ | ✗ | ✗ | ✗ |
| UnAV-100 (Geng et al., 2023) | ✓ | ✓ | ✗ | ✗ | ✓ |
| OCTAV | ✓ | ✓ | ✓ | ✓ | ✓ |

**Comparison to existing datasets** In Table 3, we compare our proposed `OCTAV` dataset to existing datasets in the audio and video domains. Most of these datasets are limited to either the video

modality (Wang et al., 2023), have missing timestamp information (Chen et al., 2023a;b; 2020), do not offer multi-turn question-answer pairs (Chen et al., 2023a;b; 2020; Geng et al., 2023) or have single event classes rather than detailed captions (Chen et al., 2020; Geng et al., 2023). OCTAV dataset addresses all the above mentioned limitations and provides a comprehensive benchmark for interwoven and fine-grained audio-visual understanding.

# 4 THE OMCAT APPROACH

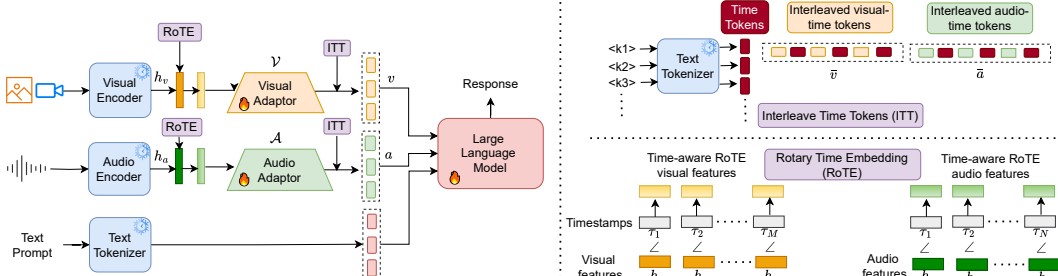

Figure 2: Overview of the OMCAT pipeline. Video frames are processed through a frozen visual encoder, while audio frames are encoded using a frozen audio encoder. Extracted features are fine-tuned through adaptor layers across all three stages. The LLM remains frozen in Stage 1 and is fine-tuned in Stages 2 and 3. The purple blocks represent time alignment modules, with only one of them activated during training. ∠ in bottom right denotes the rotation angle.

In this section, we describe our proposed OMCAT model, depicted in Figure 2. We begin by discussing the model architecture and feature extraction in Section 4.1, followed by time alignment between audio and video in Section 4.2. Next, we discuss the prompt design to query the LLM in Section 4.3 and finally, we detail the multi-stage training process of OMCAT in Section 4.4.

## 4.1 MODEL ARCHITECTURE AND FEATURE EXTRACTION

**Multi-modal Feature Extraction.** As shown in Figure 2, OMCAT uses a visual encoder, $f_v(.)$ and an audio encoder, $f_a(.)$. Given a video $V$ and an audio $A$, the encoded hidden features for the two modalities are represented as:

$$h_v = f_v(V), \quad h_a = f_a(A) \tag{1}$$

where $h_v \in \mathbb{R}^{M \times d_v}$ are the extracted features for the video modality with $M$ frames extracted uniformly from the video and $d_v$ as the hidden dimension. $M$ is 1 if the modality is image. The features for the audio modality are denoted as $h_a \in \mathbb{R}^{N \times d_a}$, where $N$ are the time windows for which the audio features are computed and $d_a$ is the hidden dimension.

**Audio-Visual Adaptors.** To map the video modality and audio modality to the text embedding space of the LLM (Chiang et al., 2023), we use two adaptor blocks: one for the video modality denoted as $\mathcal{V}(.)$ and another for the audio modality denoted as $\mathcal{A}(.)$. Essentially, the encoded hidden features are passed to the adaptors to extract token embeddings as:

$$v = \mathcal{V}(h_v), \quad a = \mathcal{A}(h_a) \tag{2}$$

These tokens are then used as prompts to the LLM along with the time representations. Following prior work (Cheng et al., 2024; Li et al., 2024), we use the fine-tuned vicuna 7B-v1.5 (Chiang et al., 2023) as our LLM to generate the final text responses. Next, we discuss how to incorporate time into our model.

## 4.2 TIME ALIGNMENT BETWEEN AUDIO AND VIDEO

Existing multimodal LLMs rely on learnable positional embeddings to encode the order of frames, but they struggle to capture the absolute time elapsed between frames and lack a fine-grained, cross-modal understanding of audio and video. We propose two strategies to encode absolute and relative temporal information on video and audio tokens, called Interleaving Time Tokens (ITT) and Rotary Time Embeddings (RoTE).

**Interleaving Time Tokens (ITT).** In this approach, we interleave time tokens with the audio and the visual features. We allocate a budget of K learnable time tokens, zero-indexed by $k_i$, and assign a time token to an audio-visual feature with the following indexing function:

$$k_i = \text{round}\left(\frac{\tau_i}{T} \cdot (K - 1)\right) \tag{3}$$

where $\tau_i$ is a continuous timestamp in seconds, $T$ is the total duration of the video or audio in seconds, and $K$ is the total number of learnable time tokens.

For a video $V$ with duration $T$ and video token embeddings $v_i$ where $i = 1 \cdots M$, each embedding is associated with a timestamp $\tau_i$ (*e.g.* 0.5 seconds, 1.4 seconds, and so forth). We first use these timestamps to obtain the discrete time tokens, then we interleave them with the visual tokens $v_i$ obtained after the visual adaptor layers. Specifically, each visual token $v_i$ corresponds to a discrete time token indexed by $k_i$, as described in Equation (3). Hence, the interleaved visual sequence is given as $\bar{v} = \{v_1, < k_1 >, v_2, < k_2 > \cdots, < v_M >, < k_M >\}$.

Similarly, for the given audio $A$ of duration $T$, we extract $N$ windows of length $w$ from the audio sequence such that for each window the time is represented as: $\tau_n = [n, n + w]$ for $n = 1, 2, \cdots, N$, where $n$ is the time in seconds. We then take the mean of the time windows, $\tau_n = \frac{n+(n+w)}{2}$. Then, we convert $\tau_n$ into discrete time token $k_n$ using Equation (3) and interleave them with the audio tokens $a$ obtained from the audio adaptor layers. Hence, the interleaved audio sequence is represented as $\bar{a} = \{a_1, < k_1 >, a_2, < k_2 > \cdots, < a_N >, < k_N >\}$. The final interleaved tokens $\bar{v}$ and $\bar{a}$ are then concatenated with the text instructions as prompts to the LLM, as shown in Figure 2 on upper top right.

**Rotary Time Embeddings (RoTE).** While we could use RoPE (Su et al., 2024) and avoid the extra context length cost introduced by ITT, RoPE would still lack the ability to capture the absolute time elapsed between frames, which is very important and crucial in scenarios with varying frame rates. To address these limitations, we propose an alternative strategy called RoTE: a modified version of RoPE, where the rotation angles are determined by absolute timestamps in seconds instead of frame indices. RoTE takes inspiration from a real clock, where each handle rotates at distinct speeds, or "frequencies". Similarly, in RoTE we rotate different dimensions in the visual and audio feature embeddings given their timestamp in seconds and the respective "frequency" of that dimension. Our results in Section 5 show that RoTE achieves performance that is superior to the baselines. A visual representation of RoTE is shown in Figure 1 on the lower right bottom.

In practice, while in rope the angle for rotation $\theta$ is defined by the temporal indexing of a token $\theta \leftarrow -i \times 2\pi$, RoTE is defined by the absolute time $\theta \leftarrow -\tau_i \times 2\pi$. These temporally enriched features are then passed to the adaptor layers $\mathcal{V}(.)$ and $\mathcal{A}(.)$ to create visual tokens $v$ and audio tokens $a$ respectively.

### 4.3 INSTRUCTION PROMPTS

In this section, we explain how video and audio tokens are combined with text prompts. The prompt format is as follows:

User: $< system\ prompt >$ Question $< vi\_start > < vi\_patch > < vi\_end > < so\_start > < so\_patch > < so\_end > < vis\_start > < vi\_patch > < so\_patch > < vis\_end >$ Assistant:

Here, $< system\ prompt >$ represents a guiding system message, following Vicuna-7B (Chiang et al., 2023). Visual and audio markers are included through tokens like $< vi\_start >/< vi\_end >$ for video and $< so\_start >/< so\_end >$ for audio. Video tokens ($< vi\_patch >$) encode visual information, and audio tokens ($< so\_patch >$) handle sound data. It is important to note that these individual video and audio markers are activated only when modality-specific data (video or audio) is present. For joint audio-video data, $< vis\_start >/< vis\_end >$ marks the boundaries, encoding both audio and video tokens, deactivating the individual representations.

### 4.4 TRAINING STRATEGY

**Stage I: Alignment Tuning Stage.** In this stage, we train the visual and audio adaptor layers and freeze the parameters of the pre-trained visual and audio encoders as well as the LLM, as shown in Figure 2. By doing so, the model can focus on learning robust features for the adaptor layers,

Table 4: List of datasets used for training OMCAT. TS indicates if timestamps are available. ST refers to single-turn question answers. MT is the version with multi-turn dialogue.

| Stage | Modality | Datasets | TS | #(Modality, Text) |
|---|---|---|---|---|
| Stage I Alignment Tuning | Image | LLaVA-Pretrain-595k (Liu et al., 2024) | ✗ | 558128 |
| | Audio | WavCaps (Mei et al., 2024) | ✗ | 403044 |
| | Video | Valley-703K (Luo et al., 2023) | ✗ | 703000 |
| | Video | VATEX (Wang et al., 2019) | ✗ | 227250 |
| | Audio-Video | VAST (Chen et al., 2023b) | ✗ | 414602 |
| | Audio-Video | VALOR (Chen et al., 2023a) | ✗ | 16109 |
| Stage II Instruction Tuning | Image | LLaVA-Tune (Liu et al., 2024) | ✗ | 624610 |
| | Audio | VGG Sound (Chen et al., 2020) | ✗ | 5157 |
| | | AudioCaps (Kim et al., 2019) | ✗ | 49838 |
| | | MusicCaps (Agostinelli et al., 2023) | ✗ | 2858 |
| | | Clotho (Drossos et al., 2020) | ✗ | 3938 |
| | | Audioset-Strong (Hershey et al., 2021) | ✓ | 431131 |
| | Video | VideoInstruct 100K (Maaz et al., 2023) | ✗ | 98145 |
| | | VideoChatGPT (Maaz et al., 2023) | ✗ | 100010 |
| | | WebVidQA (Yang et al., 2022a) | ✗ | 100000 |
| | | Valley-Instruct 65k (Luo et al., 2023) | ✗ | 64690 |
| | | VideoChat-Instruct (Li et al., 2023b) | ✗ | 6961 |
| | | Activitynet captions (Krishna et al., 2017) | ✗ | 7481 |
| | | NextQA (Xiao et al., 2021) | ✗ | 34132 |
| | | DiDeMo (Anne Hendricks et al., 2017) | ✓ | 27935 |
| | | Charades (Gao et al., 2017) | ✓ | 12408 |
| | | ActivityNet-RTL (Huang et al., 2024) | ✓ | 33557 |
| | | Youcook2 (Zhou et al., 2018) | ✓ | 8643 |
| | | ActivityNet Dense captions(Krishna et al., 2017) | ✓ | 33212 |
| | Audio-Video | Macaw Instruct (Lyu et al., 2023) | ✗ | 50656 |
| | | AVQA (Yang et al., 2022b) | ✗ | 40425 |
| | | Music-AVQA (Li et al., 2022) | ✗ | 25854 |
| | | UnAV-100 (Geng et al., 2023) | ✓ | 10358 |
| | | OCTAV-ST (Ours) | ✓ | 127507 |
| Stage III Multi-turn Instruction Tuning | Audio-Video | AVSD (Alamri et al., 2019) | ✗ | 159700 |
| | | UnAV-100-MT (Ours) | ✓ | 94916 |
| | | OCTAV-MT (Ours) | ✓ | 86000 |

which play a key role in bridging the gap between the raw audio-visual inputs and the semantic representations of the LLM.

Table 4 lists the image-text pairs (Liu et al., 2024), video-text pairs (Luo et al., 2023; Wang et al., 2019), and audio-text pairs (Mei et al., 2024) that were used to train the visual and audio adaptor layers such that the visual and audio representations are "aligned" with their corresponding textual description. In addition to these individual modalities, we also incorporate joint audio-video-text paired data (Chen et al., 2023b;a) to simultaneously train both the audio and visual adaptor layers. In total, we approximately use ∼2.3M training data. This joint training process helps the model develop a deeper understanding of the relationships between the audio and visual modalities, improving the model's ability to handle multimodal data.

**Stage II: Instruction Tuning Stage.** Following the "alignment" of modality features in Stage I, we proceed to train OMCAT using a diverse and high-quality collection of multimodal data (∼2.8M). This includes image-text, video-text, audio-text, and audio-video-text datasets that are carefully curated to prepare the model for a wide range of tasks involving video and audio. These tasks include fine-grained timestamped comprehension as well as cross-modal understanding, enabling the model to perform effectively across multiple input types. A comprehensive overview of the datasets used in this training phase is provided in Table 4. During this training stage, we freeze the parameters of both the visual and audio encoders. We only fine-tune the visual and audio adaptor layers, along with the large language model (LLM), allowing these components to be further optimized to handle multimodal tasks.

**Stage III: Multi-Turn Instruction Tuning Stage.** In the third and final stage, our main focus is to enhance the capabilities of OMCAT on multi-turn question answering in complex audio-visual scenarios. To achieve this, we fine-tune our model on multi-turn datasets, including our proposed OCTAV-MT, UnAV-100-MT, and AVSD (Alamri et al., 2019), a dataset for audio-visual dialog. Detailed statistics of these datasets are shown in Table 4. Overall, we use ∼340k training data during this stage. In this stage as well, the video encoder and the audio encoder remain frozen while we optimize the audio/video adaptor layers, along with the LLM.

## 5 EXPERIMENTS

**Datasets.** To evaluate the capabilities of OMCAT on general multimodal understanding, we evaluate our method on audio-visual understanding benchmarks. Specifically, we evaluate on the AVSD dataset (Alamri et al., 2019) which is a dataset for audio-visual scene aware dialog, Music-AVQA dataset (Li et al., 2022) that has audio-visual question answering for the music domain and AVQA dataset (Yang et al., 2022b) which has general questions about audio and visual modalities.

Furthermore, to evaluate whether OMCAT outperforms in temporal tasks, we measure the performance of our model on temporal video grounding benchmark, Charades-STA (Gao et al., 2017). This dataset is widely used in prior works (Cheng et al., 2024; Li et al., 2024; Ren et al., 2024) as a benchmark for temporal understanding.

Finally, we benchmark OMCAT on the evaluation subset of OCTAV-ST, OCTAV-MT and UnAV-100-MT datasets. These tasks require fine-grained temporal understanding, cross-correlation between the audio and visual modalities and hence are a good measure to evaluate the capabilities of OMCAT.

**Evaluation metrics.** Following prior work (Cheng et al., 2024; Li et al., 2024; Tang et al., 2024), we use GPT-4 (Achiam et al., 2023) to evaluate the answers predicted by the model by comparing against the correct answers, with a score of 0 to 5 indicating the accuracy. Besides Charades-STA where we use Recall@1 at Intersection over Union (IoU) thresholds of 0.5 and 0.7, we use the GPT accuracy everywhere else.

**Architecture.** We use the pre-trained CLIP visual encoder ViT-L/14 (Radford et al., 2021) to extract video/image features. For the audio encoder, we use the pre-trained ImageBind (Girdhar et al., 2023) model. Similar to previous work, for the video and audio adaptors, we use the Q-former which has the same architecture as the Q-Former in BLIP-2 (Li et al., 2023a). However, to maintain the temporal consistency of video and audio frames in the ITT setup, we replace the Q-Former adaptor layers with 2-layer transformer blocks with self-attention (Vaswani, 2017). During both training and inference, we sample 64 frames from the video and we extract five 3-second windows for the audio. The audio is resampled to 16KHz sampling rate and converted into spectrograms to be consistent with the input to the ImageBind model (Girdhar et al., 2023). We use 100 as the value of $K$, the learnable time tokens in Section 4.2.

**Training details.** During both the pre-training and fine-tuning stages, we train the model for one epoch on 8 NVIDIA A-100 GPUs. For the pre-training stage, we set the batch size of 64, learning rate of 1e-3 with a cosine learning decay and a warm-up period. In the fine-tuning stages, we set the batch size to 32, learning rate to 2e-5 with a cosine learning decay and a warm-up period and gradient accumulation to 2. Further details about training are given in Appendix E.

Table 5: Evaluation results for OMCAT and other state-of-the-art models on AVQA tasks (Yang et al., 2022b; Alamri et al., 2019; Li et al., 2022), Charades-STA (Gao et al., 2017) and our proposed OCTAV-ST dataset. While † describes results from models fine-tuned on the training set of those datasets, results in parentheses are zero-shot.

| Method | Time | Accuracy | | | R@1(IoU=0.5) | R@1(IoU=0.7) | Accuracy | |
| | | | | | | | OCTAV-ST Youcook2 | OCTAV-ST ActivityNet |
| | | AVSD | Music-AVQA | AVQA | Charades-STA | | | |
| PandaGPT (Su et al., 2023) | ✗ | 26.1† | 33.7 | 79.8† | - | - | - | - |
| Video LLaMA (Cheng et al., 2024) | ✗ | 36.7† | 36.6 | 81.0† | 3.8 | 0.9 | - | - |
| MacawLLM (Lyu et al., 2023) | ✗ | 34.3† | 31.8 | 78.7† | - | - | - | - |
| AVLLM (Shu et al., 2023) | ✗ | 52.6† | 45.2 | - | - | - | - | - |
| AVicuna (Tang et al., 2024) | ✓ | 53.1† | 49.6 | - | - | - | - | - |
| Video LLaMA 2 (Zhang et al., 2023) | ✗ | 53.3† | 73.6† | - | - | - | 9.14 | 10.55 |
| GroundingGPT (Li et al., 2024) | ✓ | - | - | - | 29.6† | 11.9† | 1.20†(3.87) | 1.57†(7.6) |
| OMCAT (RoTE) | ✓ | 49.4† | 73.8†(51.2) | 90.2† | 32.3† | 15.9† | 16.9†(9.9) | 19.0†(11.2) |

### 5.1 QUANTITATIVE RESULTS

**Comparison to state-of-the-art.** We follow previous work (Cheng et al., 2024; Zhang et al., 2023; Shu et al., 2023) to evaluate OMCAT on three audio-video understanding benchmarks. Based on the GPT-assisted evaluation scores in Table 5, our model surpasses the most recent and relevant models on all benchmarks. While on Music-AVQA we achieve 51.2% accuracy in the zero-shot setting and 73.8% in the fine-tuned setting, outperforming SOTA models, on AVQA dataset we significantly outperform other models. We believe our competitive but relatively lower scores on AVSD comes from a difference in data distribution during the final training stage.

To evaluate temporal understanding in videos, we evaluate `OMCAT` Charades-STA, an established benchmark for this task. We outperform GroundingGPT (Li et al., 2024) on Recall@1 at IoU threshold of 0.5 and 0.7. This result shows that our method can also perform temporal understanding in the video domain.

Finally, we present results on the single-turn version of our proposed `OCTAV` benchmark, `OCTAV-ST`. We evaluated VideoLLaMA2 (Zhang et al., 2023) in a zero-shot setting on this dataset and fine-tuned GroundingGPT (Li et al., 2024) on the `OCTAV-ST` training set for a fair comparison. As shown in Table 5, our method outperforms all the above two methods in both the zero-shot (results in parantheses) and fine-tuned settings. These results confirm `OMCAT`'s ability to jointly learn cross-modal and temporal understanding from both video and audio data.

Table 6: Results of different variations of `OMCAT` (RoPE, ITT and `RoTE`) on the `OCTAV-MT` benchmark and the UnAV-100-MT dataset.

| Method | Accuracy | | |
| --- | --- | --- | --- |
| | `OCTAV-MT`-**Youcook2** | `OCTAV-MT`-**ActivityNet** | **UnAV-100-MT** |
| GroundingGPT (Li et al., 2024) | 0.13 | 0.07 | 13.2 |
| `OMCAT` (RoPE) | 3.3 | 2.4 | 15.7 |
| `OMCAT` (ITT) | 3.1 | 4.1 | 16.6 |
| `OMCAT` (`RoTE`) | **3.7** | **5.6** | **19.9** |

**Comparison on the `OCTAV-MT` benchmark.** In Table 6, we highlight the performance of `OMCAT` on the `OCTAV-MT` benchmark, which involves multi-turn question-answer pairs for videos with multiple sound events. All models in Table 6 are fine-tuned on the proposed `OCTAV-MT` benchmark. Our model, `OMCAT` with `RoTE`, significantly outperforms the baselines—ITT, RoPE, and GroundingGPT (Li et al., 2024)—on this dataset. Moreover, it achieves substantial performance gains on the UnAV-100-MT dataset, a dataset with in-the-wild/natural audio-visual events (*e.g.* Figure 7).

`OMCAT` with `RoTE` efficiently integrates time representations with minimal computational cost, ensuring precise cross-modal alignment between audio and video. While these improvements over the baselines are considerable, there is still ample room for further enhancement in this area. The `OCTAV-MT` benchmark paves the way for the development of advanced multimodal models with stronger cross-modal grounding capabilities.

Table 7: Effect of applying various time embeddings–RoPE, ITT and `RoTE` to `OMCAT` on all benchmarks.

| Time Encoding | Accuracy | | | R@1(IoU=0.5) | R@1(IoU=0.7) | Accuracy | |
| --- | --- | --- | --- | --- | --- | --- | --- |
| | AVSD | Music-AVQA | AVQA | \<Charades-STA\> | | `OCTAV-ST`-Youcook2 | `OCTAV-ST`-ActivityNet |
| RoPE | 45.9 | 71.2 | 88.2 | 30.7 | 16.1 | 13.3 | 16.5 |
| ITT | 47.3 | 69.7 | 82.1 | **32.5** | **16.7** | 16.5 | **19.2** |
| RoTE | 49.4 | 73.8 | 90.2 | 32.3 | 15.9 | 16.9 | 19.0 |

Table 8: Effect of alignment tuning data on the overall performance. LP denotes LLaVA-Pretrain-595k (Liu et al., 2024), WC denotes WavCaps (Mei et al., 2024) and, V denotes Valley-703K (Luo et al., 2023).

| Ablation | Music-AVQA | Charades-STA (R@1,IoU-0.5) | `OCTAV-ST`-Youcook2 |
| --- | --- | --- | --- |
| `OMCAT` w/ only LP,WC,V | 50.6 | 26.9 | 4.97 |
| Ours | **51.2** | **32.3** | **16.9** |

## 5.2 ABLATION STUDY

**How does time embedding affect `OMCAT`?** In Table 7, we evaluate three different time embedding approaches, including RoPE (Su et al., 2024), and our proposed approaches ITT and `RoTE`. On the AVQA benchmark, `RoTE` consistently outperforms the baselines by a large margin, demonstrating its strong capability not only on temporal and cross-modal tasks but also in handling coarse-grained question answering.

For the temporal understanding task on Charades-STA, ITT performs slightly better than `RoTE` at both IoU thresholds (0.5 and 0.7). On the `OCTAV-ST` benchmark, YouCook2 and ActivityNet, ITT and `RoTE` show nearly equivalent performance. We believe ITT's competitive results stem from its

explicit time embedding through time tokens. However, given ITT's increased context length and its weaker performance on AVQA tasks, RoTE is the more effective and efficient choice overall.

**What is the effect of pre-training data on OMCAT?** Furthermore, we investigate the impact of pre-training data on the final model performance, particularly during the alignment tuning stage (Stage I). This stage is crucial for establishing the model's capacity to "align" information across different modalities, such as image, video, and audio, with text. To examine the role of joint multimodal data, we conduct an ablation study where we modify the training data by excluding the audio-video-text paired data (Chen et al., 2023b;a) while retaining image-text (Liu et al., 2024), video-text (Luo et al., 2023; Wang et al., 2019), and audio-text pairs (Mei et al., 2024).

Our results in Table 8 indicate a noticeable decline in performance across all tasks when the model is trained without audio-video-text data. This demonstrates the critical importance of joint multimodal data in achieving robust cross-modal alignment. We hypothesize that without data that directly links audio, video, and text, the model struggles to accurately capture the intricate relationships between these modalities, leading to suboptimal performance in tasks requiring fine-grained multimodal understanding.

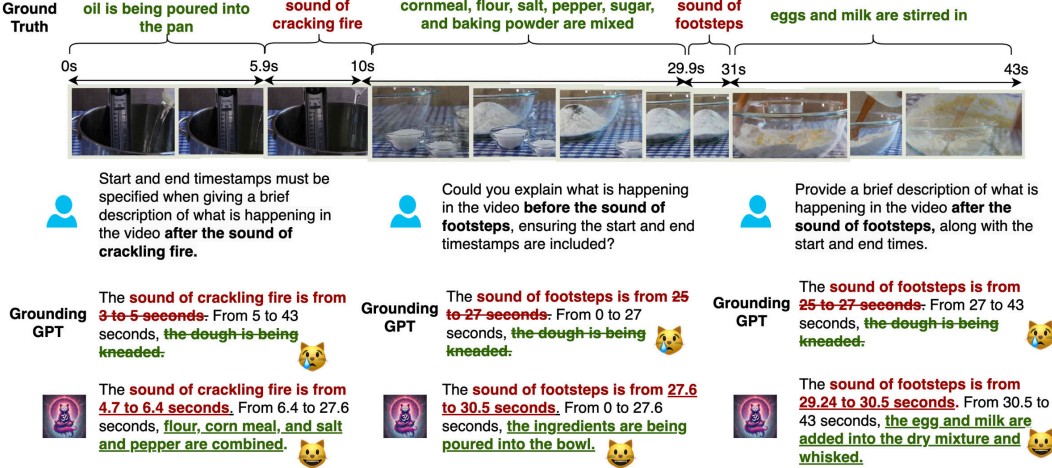

Figure 3: Qualitative comparison of OMCAT with GroundingGPT on the OCTAV-MT dataset.

## 5.3 QUALITATIVE RESULTS

In Figure 3, we showcase the qualitative performance of our method on the YouCook2 subset of the OCTAV-MT benchmark. GroundingGPT inaccurately predicts a uniform activity of *dough being kneaded*, failing to capture the nuanced transitions in events triggered by sound cues. In contrast, our model successfully isolates specific events and accurately associates them with their corresponding timestamps based on the sound events. For instance, our model correctly identifies the activity following the *sound of cracking fire* (around 6.4 to 27.6 seconds), predicting that *flour, cornmeal, and salt and pepper are combined*. This aligns closely with the ground truth, which describes the activity as *cornmeal, flour, salt, pepper, sugar, and baking powder being mixed*. While OMCAT omits some ingredients, it still recognizes the correct activity—unlike GroundingGPT, which mistakenly predicts *dough being kneaded*.

Similarly, OMCAT accurately predicts that *egg and milk are added into the dry mixture and whisked* following the *sound of footsteps* (from 29.2 to 30.5 seconds). However, when asked what occurs before the sound of footsteps, the model correctly predicts the activity as *ingredients being mixed in the bowl*, though the prediction does not perfectly match the ground truth.

## 6 CONCLUSION

In this paper, we addressed the limitations of multimodal large language models in fine-grained, cross-modal temporal understanding by introducing the OCTAV dataset and the OMCAT model. OCTAV focuses on event transitions across audio and video, promoting deeper temporal alignment and cross-modal understanding. OMCAT, enhanced with RoTE embeddings, effectively grounds temporal information across modalities, leading to superior performance on Audio-Visual Question Answering (AVQA) tasks and the OCTAV benchmark. Our approach sets a new standard for multimodal AI, advancing cross-modal and temporal reasoning capabilities for future research.

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

APPENDIX

## A    DEMO PAGE LINK

The link to our demo page is `https://om-cat.github.io/`.

## B    ALGORITHM FOR ROTARY TIME EMBEDDINGS (ROTE)

Below, we present the algorithm for applying RoTE to the visual feature embeddings, $h_v$. The same algorithm applies to audio features, $h_a$.

---

**Algorithm 1** Rotary Time Embeddings (RoTE)

---

**Require:** $h_{v_x}$: List of x-coordinates for visual features, $h_{v_y}$: List of y-coordinates for visual features, $\tau$: timestamp for the video frames, $d$: Embedding dimension, $N$: number of video frames
**Ensure:** Rotated coordinates $(h_{v_{x_{rot}}}, h_{v_{y_{rot}}})$
 1: $freqs \leftarrow linspace(0, 1, d/2)$            ▷ *Frequency adjustments based on dimension*
 2: **for** $i = 1$ *to* $N$ **do**
 3:     $\theta \leftarrow -\tau_i \times 2\pi$                            ▷ *Angle for rotation*
 4:     $rot\_angle \leftarrow \theta \times freqs[-1]$         ▷ *Rotation using the highest frequency*
 5:     $h_{v_{x_{rot}}}[i] \leftarrow h_{v_x}[i] \times \cos(rot\_angle) - h_{v_y}[i] \times \sin(rot\_angle)$
 6:     $h_{v_{y_{rot}}}[i] \leftarrow h_{v_x}[i] \times \sin(rot\_angle) + h_{v_y}[i] \times \cos(rot\_angle)$
 7: **end for**
 8: **return** $(h_{v_{x_{rot}}}, h_{v_{y_{rot}}})$

---

## C    PROMPTS FOR GENERATING OCTAV DATASET

In this section, we discuss further details about generating our proposed dataset.

### C.1    PROMPTS FOR OCTAV-ST DATASET

Below we show the prompts used to generate question-answer pairs for the video conditioned on a single audio event *i.e.* OCTAV-ST dataset.

You are an AI assistant that can analyze a video. You receive timestamped video and audio captions with start time and end times describing the video you are observing. Based on these audio and video captions, create 2 question and answer pairs where a question is asked by the person (the user) and the answer is given by you (the assistant) about the events in the video/audio. Here are some additional requirements about the generated question-answer pairs:
1. The question asked by the user should be from the audio caption and the answer given by the assistant should be from the video caption before or after that timestamp in question.
2. Only describe what you are certain about, and avoid providing descriptions that maybe ambiguous or inaccurate.
4. The number of words in the answer should not exceed 100 words. Keep it as concise as possible. You do not need to include everything in the answer.
Include timestamp information in the answers.

Example 1:
Timestamped video and audio captions:
"video caption 1": season the chicken on both sides with salt and pepper then cut it into pieces from 0.0 to 18, "video caption 2": put the chicken pieces to a boiling pot of water cover it and let it cook from 20 to 22, "audio caption": There is a sound of Trumpet from 18 to 20.

QA:
User: What is happening in the video before the sound of trumpet? Assistant: The sound of trumpet is from [18.0, 20.0]. From [0.0, 18.0], the chicken is seasoned on both sides with salt and pepper then cut it into pieces.
User: What is happening in the video after the sound of trumpet? Assistant: The sound of trumpet is from [18.0, 20.0]. From [20.0, 22.0], the chicken pieces are put to a boiling pot of water, covered and then cooked.

Based on the example above, design 2 question and answer pairs between the user and assistant for the example given below.
Format each QA pair in a single line as a JSON dictionary (key "user" for question, and "assistant" for answer).

## C.2 PROMPTS FOR OCTAV-MT DATASET

Below we show the prompts used to generate question-answer pairs for the video conditioned on two audio events *i.e.* OCTAV-MT dataset.

You are an AI assistant that can analyze a video. You receive timestamped video and audio captions with start time and end times describing the video you are observing. Based on these audio and video captions, create 4 question and answer pairs where a question is asked by the person (the user) and the answer is given by you (the assistant) about the events in the video/audio. You can ask clarification questions if the question asked by the user is not clear. Here are some additional requirements about the generated question-answer pairs:

1. The question asked by the user can be from the audio caption or the video caption and the answer given by the assistant should be from the video caption before or after that timestamp in question.

2. Only describe what you are certain about, and avoid providing descriptions that maybe ambiguous or inaccurate.

4. The number of words in the answer should not exceed 100 words. Keep it as concise as possible. You do not need to include everything in the answer.

Include timestamp information in the answers.

Example 1:

Timestamped video and audio captions:

"video caption 1": season the chicken on both sides with salt and pepper then cut it into pieces from 0.0 to 18, "video caption 2": put the chicken pieces to a boiling pot of water cover it and let it cook from 20 to 22, "video caption 3": chop celery to small pieces chop cheese to cubes and chop ham also to the same size from 26 to 50, "audio caption 1": There is a sound of laugh from 18 to 20, "audio caption 2": There is a sound of laugh from 22 to 26.

QA:

User: What is happening in the video after the sound of laugh? Answer with start and end timestamps.

Assistant: There are two sounds of laugh, one from [18.0, 20.0] and the other one from [22.0, 26.0]. Which laugh are you referring to?

User: I am referring to the laugh that happens after the chicken pieces are out to a boiling pot of water.

Assistant: Okay, so the laugh from [22.0, 26.0]. After this sound of laugh from [26.0, 50.0], celery is chopped to small pieces, cheese is chopped to cubes and ham is chopped also to the same size.

User: Thanks, what is happening in the video after the chicken is seasoned on both sides with salt and pepper. Answer with start and end timestamps.

Assistant: There is a sound of laugh from [18.0, 20.0] and from [20.0, 22.0], the chicken pieces are put ot a boiling pot of water, covered and cooked.

User: Thanks, what is happening in the video after the sound of bird chirping? Answer with start and end timestamps.

Assistant: Sorry, there is no sound of bird chirping.

Based on the above examples, design 4 question and answer pairs between the user and assistant for the example given below.

Format each QA pair in a single line as a JSON dictionary (key "user" for question, and "assistant" for answer, wrapped with  and ).

## C.3 LIST OF INSTRUCTIONS

Below, we show the diverse set of instructions that we use to replace the common instruction *What is happening in the video* generated by the GPT model. The *eventname* below is replaced by the anchored query such as *after the sound of bird chirping*.

Start and end timestamps should be included while describing what *eventname* is.

Please include the start and end time when briefly describing what *eventname* entails.

Start and end timestamps are required while providing a brief description of what *eventname* involves.

Include the exact start and end times when describing what *eventname* refers to.

Ensure to mention the start and end timestamps when explaining what *eventname* covers.

With the start and end times, please provide a brief explanation of what *eventname* is.

Start and end timestamps should be given alongside a description of what *eventname* involves.

When describing what *eventname* is, include the exact start and end time information.

Include start and end time details when summarizing what *eventname* entails.

Start and end timestamps must be specified when giving a brief description of what *eventname* refers to.

Describe what *eventname* is with start and end timestamps.

Please briefly describe what *eventname* entails, including its exact start and end timestamps.

Provide a brief description of what *eventname* includes, along with the start and end times.

Give a short description of what *eventname* is, including the precise start and end time details.

Briefly explain what *eventname* involves, including its start and end timestamps.

Please summarize what *eventname* covers, specifying the start and end timestamps.

Give a brief explanation of what *eventname* is, making sure to include both the start and end times.

Could you describe what *eventname* refers to, including the exact start and end times?

Please provide a concise overview of what *eventname* involves, along with start and end time details.

Could you explain what *eventname* is, ensuring the start and end timestamps are included?

## C.4 SOUND EVENTS

In this section, we provide details about the datasets we used for adding sound to the curated chunked videos as discussed in Section 3. Specifically, we use Urban Sound 8K (Salamon et al., 2014), ESC-50 (Piczak, 2015), FSD50K (Fonseca et al., 2021) and NonSpeech7K (Rashid et al., 2023) datasets.

Urban Sound 8K (Salamon et al., 2014) is an audio dataset that contains urban sounds from 10 classes: air conditioner, car horn, children playing, dog bark, drilling, engine idling, gun shot, jackhammer, siren, and street music.

The ESC-50 dataset (Piczak, 2015) consists of 5-second-long recordings organized into 50 semantical classes that can be categorized into 5 major categories of animals, natural soundscapes & water sounds, human and non-speech sounds, interior/domestic sounds and exterior/urban noises.

FSD50K (Fonseca et al., 2021) has 200 sound categories mainly produced by physical sound sources and production mechanisms, including human sounds, sounds of things, animals, natural sounds, musical instruments and more.

Nonspeech7k (Rashid et al., 2023) contains a diverse set of human non-speech sounds, such as the sounds of breathing, coughing, crying, laughing, screaming, sneezing, and yawning.

## D    EXAMPLES FROM THE `OCTAV` DATASET

In Figure 4, we show examples from the `OCTAV-ST` dataset. The top part of the figure shows an example from the ActivityNet subset and the bottom part shows an example from the Youcook2 subset of the dataset. These examples give an overview of how different event transitions are interwoven seamlessly with an audio event.

In Figure 5 and Figure 6, we show examples from the ActivityNet subset and the Youcook2 subset of the `OCTAV-MT` dataset respectively. These examples show the anchoring of transitioning video events on multiple sound events.

In Figure 7, we show an example from the UnAV-100-MT dataset, which is the multi-turn version of the UnAV-100 dataset (Geng et al., 2023). We convert the audio-visual timestamped annotations from the UnAV-100 dataset into multi-turn question answers as shown in this example. This dataset acts as a benchmark for a real time setting of audio-visual scenarios.

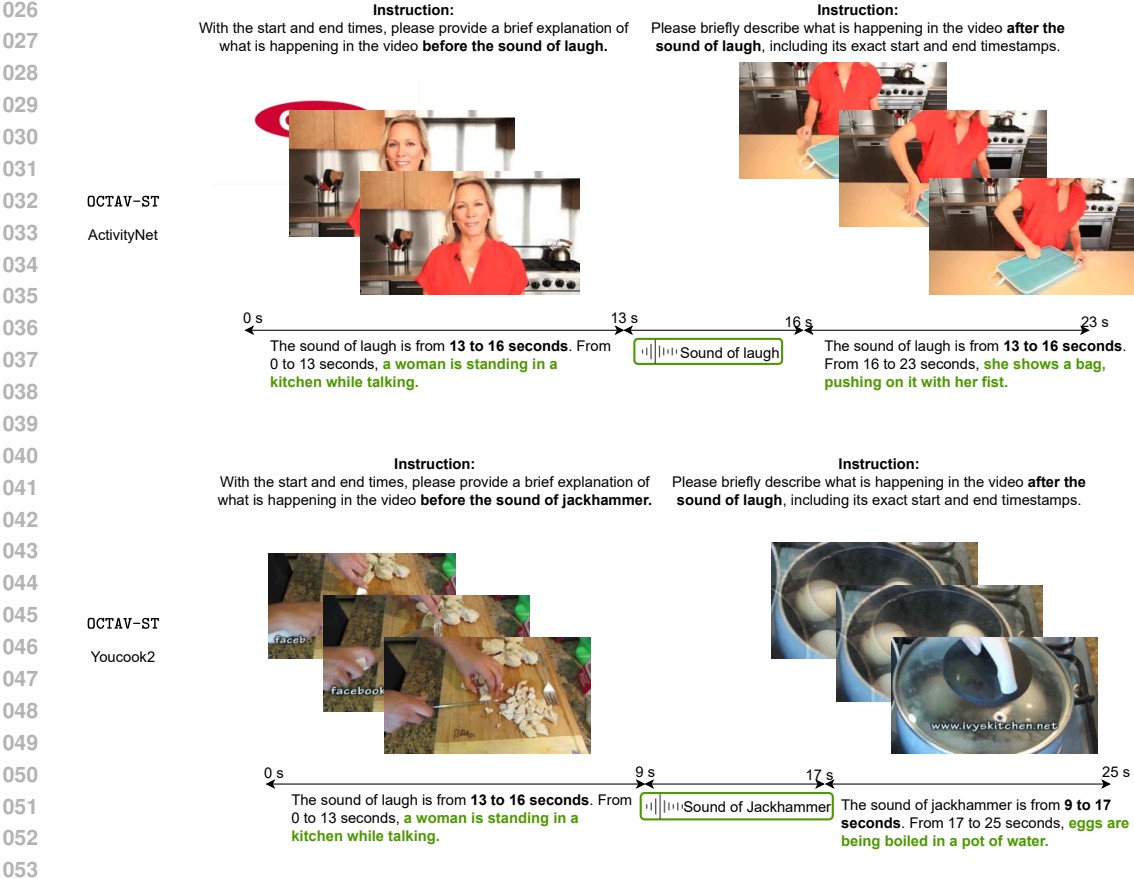

Figure 4: Question-answer pairs from the proposed OCTAV-ST dataset.

Figure 5: Multi-turn dialogue examples from the ActivityNet subset of our OCTAV-MT dataset.

OCTAV-MT

Youcook2

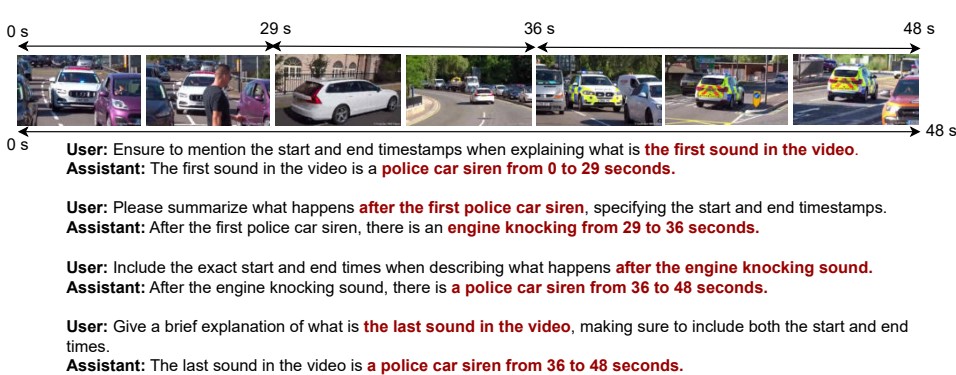

0 s                                                                                                50 s

**User:** Please briefly describe what is happening in the video **after the sound of street music**, including its exact start and end timestamps.
**Assistant:** The sound of street music is from **15 to 23 seconds**. From 23 to 40 seconds, **carrots are grated and transferred to the big bowl.**

**User:** Briefly explain what is happening in the video **before the sound of gun shot**, including its start and end timestamps.
**Assistant:** The sound of gun shot is from **40 to 41 seconds**. From 23 to 40 seconds, **carrots are grated and transferred to the big bowl.**

**User:** Start and end timestamps should be given alongside a description of what is happening in the video **after the sound of gun shot.**
**Assistant:** The sound of gun shot is from **40 to 41 seconds**. From 41 to 50 seconds, **diced onion and garlic are added to the bowl.**

**User:** When describing what is happening in the video **before the sound of street music**, include the exact start and end time information.
**Assistant:** The sound of street music is from **15 to 23 seconds**. From 0 to 15 seconds, **purple cabbage is processed and transferred to the big bowl.**

Figure 6: Multi-turn dialogue examples from the Youcook2 subset of our OCTAV-MT dataset.

UnAV-100-MT

0 s                    29 s                    36 s                    48 s

0 s                                                                                  48 s

**User:** Ensure to mention the start and end timestamps when explaining what is **the first sound in the video**.
**Assistant:** The first sound in the video is a **police car siren from 0 to 29 seconds.**

**User:** Please summarize what happens **after the first police car siren**, specifying the start and end timestamps.
**Assistant:** After the first police car siren, there is an **engine knocking from 29 to 36 seconds.**

**User:** Include the exact start and end times when describing what happens **after the engine knocking sound.**
**Assistant:** After the engine knocking sound, there is **a police car siren from 36 to 48 seconds.**

**User:** Give a brief explanation of what is **the last sound in the video**, making sure to include both the start and end times.
**Assistant:** The last sound in the video is **a police car siren from 36 to 48 seconds.**

Figure 7: Multi-turn dialogue examples from the UnAV-100-MT dataset.

## E  TRAINING DETAILS

The training settings for our model are spread across three stages, each with specific configurations.

Stage 1 utilizes a batch size of 64 and a learning rate of 1e-3, employing a cosine decay learning schedule. The warm-up ratio is set at 0.03, with no weight decay applied. This stage runs for 1 epoch with gradient accumulation of 1. Additionally, it employs the ZeRO2 optimization strategy in DeepSpeed (Rasley et al., 2020) and utilizes 8 A100 GPUs.

Stage 2 has a smaller batch size of 32 and reduces the learning rate to 2e-5. It follows the same warm-up ratio of 0.03 and applies no weight decay. Like Stage 1, this stage runs for 1 epoch but increases gradient accumulation to 2. The same DeepSpeed optimization and GPU configuration are used.

Stage 3 mirrors the settings of Stage 2, with a batch size of 32, a learning rate of 2e-5, and a warm-up ratio of 0.03. It also has no weight decay and runs for 1 epoch with gradient accumulation set to 2. We use the same DeepSpeed optimization and 8 A100 GPUs like the previous stages.

## F  MORE RESULTS

In Table 9, we showcase the zero-shot performance of our proposed OMCAT model on the video understanding benchmarks MSRVTT-QA (Xu et al., 2016), MSVD-QA (Chen & Dolan, 2011), and ActivityNet-QA (Yu et al., 2019). Although our model's performance falls short compared to Video

LLaMA 2 (Cheng et al., 2024) and AVicuna (Tang et al., 2024), it remains competitive with other models in the field (Li et al., 2024; Zhang et al., 2023; Li et al., 2023b). We attribute AVicuna's higher performance to its instruction tuning with ActivityNet captions (Krishna et al., 2017) and its specialization in video understanding during the final training stage. Similarly, Video LLaMA 2 (Cheng et al., 2024) is also an expert model, having been trained on a significantly larger video-text dataset throughout all training phases, unlike `OMCAT`.

We further assess our method's effectiveness in audio understanding by evaluating it on the Clotho-AQA (Lipping et al., 2022) dataset, where `OMCAT` achieves a score of 54.3% in audio question answering. In comparison, the audio expert model Qwen-Audio (Chu et al., 2023) scores 57.9%, while Video LLaMA 2 reaches 59.7%. Our model demonstrates competitive performance on this benchmark; however, we believe that the extensive audio-text training data utilized by these two models contributes to their superior results. Moreover, we use Imagebind (Girdhar et al., 2023) as our audio encoder whereas these models use a far more superior audio encoder pre-trained on a large-scale audio-text data unlike Imagebind (Girdhar et al., 2023). It is worth noting that this aspect was beyond the scope of our work, which primarily focuses on temporal and cross-modal understanding of audio and video.

Table 9: Performance comparison on video understanding benchmarks. † means specialized model and ∗ means trained on a much larger dataset.

| Method | Modality | MSRVTT-QA | MSVD-QA | ActivityNet-QA |
|---|---|---|---|---|
| VideoChat (Li et al., 2023b) | Video | 45.0 | 56.3 | 26.5 |
| Video-ChatGPT (Maaz et al., 2023) | Video | 49.3 | 64.9 | 35.2 |
| Valley (Luo et al., 2023) | Video | 45.7 | 65.4 | 42.9 |
| Video-LLaMA (Zhang et al., 2023) | Video | 29.6 | 51.6 | 12.4 |
| PandaGPT (Su et al., 2023) | Video, Audio | 23.7 | 46.7 | 11.2 |
| MacawLLM (Lyu et al., 2023) | Video, Audio | 25.5 | 42.1 | 14.5 |
| AVLLM (Shu et al., 2023) | Video, Audio | 53.7 | 67.3 | 47.2 |
| GroundingGPT (Li et al., 2024) | Video, Audio | 51.6 | 67.8 | 44.7 |
| AVicuna† (Tang et al., 2024) | Video, Audio | **59.7** | 70.2 | **53.0** |
| Video LLaMA 2∗ (Cheng et al., 2024) | Video, Audio | 53.9 | **71.7** | 49.9 |
| `OMCAT` (RoPE (Su et al., 2024)) | Video, Audio | 49.3 | 63.2 | 41.9 |
| `OMCAT` (ITT) | Video, Audio | 51.1 | 65.1 | 43.9 |
| `OMCAT` (RoTE) | Video, Audio | 51.2 | 67.8 | 46.6 |

## G LIMITATIONS AND FUTURE WORK

Here, we outline some limitations that are important considerations for future work.

First, the `OCTAV` dataset consists of sounds that are non-overlapping and distinct, which simplifies the learning and classification process. However, in real-life scenarios, sound events often overlap, occur simultaneously, and can be highly ambiguous. This makes sound detection and classification far more complex. Thus, a natural extension of our work would be to incorporate sound data that reflects more in-the-wild conditions, where sounds are less controlled, overlap frequently, and can exhibit high variability in intensity and duration. Adapting the dataset to represent such real-world complexities will enhance the robustness and applicability of the model in practical applications.

Second, our proposed `OMCAT` model employs the CLIP visual encoder (Radford et al., 2021) as the video encoder, which focuses on frame-based visual representations. While CLIP has demonstrated strong capabilities in multimodal learning, it lacks explicit modeling of temporal dynamics between video frames. Given that many real-world events are temporally dependent—especially in video sequences—using a video-based encoder that captures temporal consistency, such as those designed for action recognition (Ren et al., 2024), would likely result in more accurate and nuanced representations of events. In future work, we aim to explore alternative video encoders that model temporal aspects of video more effectively, enabling better alignment between the visual and audio modalities in complex, dynamic environments. This could lead to more sophisticated models capable of handling temporal dependencies and multi-event interactions in both visual and audio data.

Third, currently the dataset consists of short-length videos (∼30-40 seconds), extending the dataset to long videos would be extremely beneficial for practical applications. Longer videos would provide more comprehensive context, allowing models to better capture temporal dependencies, complex patterns, and interactions that unfold over extended periods. Moreover, long-duration videos would

enable more robust testing and evaluation in real-world scenarios, where short clips often fail to represent the full dynamics of real-time events. Expanding the dataset in this way would lead to more accurate models and improve their generalizability across a broader range of applications.

