# OpenReview forum: "OMCAT: Omni Context Aware Transformer"
_ICLR.cc/2025/Conference — Submitted to ICLR 2025_

### Official Review · Reviewer_Q1EU · 2024-10-28

**Soundness:** 2
**Presentation:** 3
**Contribution:** 3
**Rating:** 5
**Confidence:** 4

**Summary:**

The paper introduces OMCAT, a model aimed at advancing cross-modal temporal understanding by leveraging both audio and visual inputs. To support this objective, the authors present OCTAV, a dataset specifically designed to enable detailed event correlation between audio and video streams. The model employs a three-stage training approach and incorporates Rotary Time Embeddings (RoTE) to enhance temporal grounding. Experimental results showcase the model's effectiveness on Audio-Visual Question Answering (AVQA) tasks, as well as its adequate performance on the newly introduced OCTAV benchmark.

**Strengths:**

The authors address a well-defined gap in existing multimodal models—fine-grained temporal reasoning between audio and video inputs. They contribute both a new dataset (OCTAV) and a novel model (OMCAT), advancing the field significantly.

OCTAV fills the need for datasets that handle fine-grained temporal and cross-modal alignment. The synthetic generation strategy for question-answer pairs is innovative and provides a valuable resource for future research.

The experimental evaluation is thorough, covering multiple benchmarks and including both quantitative and qualitative analyses. The performance gains on AVQA and temporal grounding tasks are compelling evidence of the model's efficacy.

**Weaknesses:**

The use of GPT-4 for evaluation introduces an element of subjectivity, especially in grading responses on a 0-5 scale. Although this approach is common, the absence of standardized quantitative metrics could complicate comparisons with future studies. Additionally, human evaluation is recommended to achieve a more reliable and robust assessment.

Given that the dataset is primarily constructed using GPT, I am concerned about its quality control. It would be beneficial to incorporate human annotation or validation to ensure greater accuracy and reliability.

While the paper compares OMCAT to several state-of-the-art models, it would be beneficial to include more baselines from diverse modalities to fully illustrate its strengths and weaknesses across different scenarios.

**Questions:**

The paper relies heavily on GPT-4 for scoring responses with a subjective grading scale. How consistent and reliable is this approach across different response types? Did the authors test the robustness of the GPT-4 evaluation process, such as through inter-rater reliability tests or additional scoring methods?

Can the authors elaborate on the quality control measures applied to ensure the reliability and accuracy of the synthetically generated OCTAV dataset? Given that synthetic data may sometimes lack the subtle variations of real-world data, how did the authors verify that OCTAV’s question-answer pairs and temporal annotations closely reflect realistic audio-visual interactions? Additionally, are there any known limitations in OCTAV’s data quality that might impact OMCAT’s generalizability?

---

> ### Author Response · Authors · 2024-11-21
> **Response to Reviewer Q1EU**
>
> We thank the reviewer for their positive feedback and comments that will greatly help in improving this work. Below, we address all the concerns raised.
>
> >**The use of GPT-4 for evaluation introduces an element of subjectivity, especially in grading responses on a 0-5 scale. Although this approach is common, the absence of standardized quantitative metrics could complicate comparisons with future studies. Additionally, human evaluation is recommended to achieve a more reliable and robust assessment.**
>
> A key advantage of using GPT-4 for evaluation over traditional metrics is its ability to handle open-ended answers, where a response may be correct but not identical to the ground truth. While human evaluation is the gold standard, it is highly expensive and time-intensive. Recent works [1] have shown reliability of GPT-4 for evaluation under different scenarios and it has proven robust to response types, prompts, style changes and iterations.  Furthermore, evaluating responses with GPT-4 has been a very effective tool for reinforcement learning with AI feedback to align model responses closer to human responses [2,3].
>
> >**Given that the dataset is primarily constructed using GPT, I am concerned about its quality control. It would be beneficial to incorporate human annotation or validation to ensure greater accuracy and reliability.**
>
> Thanks for raising this. For assessing the quality of our generated dataset, we trained OMCAT without the OCTAV dataset and compared the results of this method on the final model trained with OCTAV. As shown in the results below, training with the OCTAV dataset leads to a gain in performance not only on the OCTAV benchmark but also across temporal understanding tasks (e.g. Charades-STA).
> | Model                        | Charades-STA  | OCTAV-ST-Youcook2 |
> |-------------------------|------------------ |------------------------- |
> | OMCAT w/o OCTAV  |           30.8        |                 9.9              |
> | OMCAT w/ OCTAV    |           32.3        |                 16.9            |
>
> Moreover, the QA task in OCTAV is closely tied to the ground-truth video and audio time-stamped annotations, both of which are human annotated, providing a solid foundation for quality control.
>
> Furthermore, the GPT-4-assisted generation heavily relies on prompt tuning and in-context examples (see Appendix B.1 and B.2). This approach effectively minimizes the risk of errors or hallucinations in question-answer generation, delivering reliable quality without the need for extensive human validation.
>
> >**While the paper compares OMCAT to several state-of-the-art models, it would be beneficial to include more baselines from diverse modalities to fully illustrate its strengths and weaknesses across different scenarios.**
>
> In Table 9 (in the Appendix) we compare OMCAT’s video understanding ability on three benchmark datasets MSVD-QA, MSRVTT-QA and ActivityNet QA. Many multimodal LLMs, such as Video LLaMA 2, leverage vast amounts of video-text paired data (around 12 million pairs), which boosts their video understanding capabilities. In contrast, OMCAT is trained on a much smaller subset of video-text data (about 2 million pairs), which naturally limits its video understanding performance. Despite this, OMCAT demonstrates competitive results in video understanding tasks.
>
> Furthermore, we evaluate OMCAT on the audio question-answering task using the Clotho-AQA dataset [4], where it achieved an accuracy of 57.4%, closely matching the 57.9% achieved by Qwen-7B Audio [5], a model specifically designed for audio only understanding tasks. This demonstrates that OMCAT excels not only in cross-modal and temporal tasks but also sustains strong performance in modality-specific audio understanding. We will add these results in the paper.
>
> >**The paper relies heavily on GPT-4 for scoring responses with a subjective grading scale. How consistent and reliable is this approach across different response types? Did the authors test the robustness of the GPT-4 evaluation process, such as through inter-rater reliability tests or additional scoring methods?**
>
> Following previous work, we use the GPT-4 based evaluation for consistency and fair comparison. Recent works [1] have shown reliability of GPT-4 for evaluation under different scenarios and it has proven robust to response types, prompts, style changes and iterations.

---

> > ### Author Response · Authors · 2024-11-21
> > **Response to Reviewer Q1EU**
> >
> > >**Given that synthetic data may sometimes lack the subtle variations of real-world data, how did the authors verify that OCTAV’s question-answer pairs and temporal annotations closely reflect realistic audio-visual interactions?**
> >
> > The OCTAV dataset contains videos collected from open-source datasets like YouCook2 and ActivityNet, which are sourced from the web and depict real-world scenarios. The video content hence is quite realistic, interleaved with natural sounds in between the video events.
> >
> > Real-world applications can be a bit ambiguous, and hence we address this concern by:
> >
> > i) **providing a benchmark on "natural" videos**. We evaluated the model on the UnAV-100 dataset (refer to Table 6, last row) in the main paper, which features real-time audio-visual events. Although the annotation style in UnAV-100 differs somewhat from our proposed OCTAV dataset, it still offers a robust benchmark for testing cross-modal and temporal understanding. As demonstrated in the example provided in the appendix (Figure 7), UnAV-100 dataset has complex, dynamic, real-world scenarios, making it a good testbed for real-world applications.
> >
> > ii) **providing an example with a movie trailer**. We updated the section “Temporal Question Answering on natural videos” in the demo website: https://om-cat.github.io/ to capture this. Specifically, the first two examples show the capability of OMCAT model to identify musical capabilities on movie trailers from Whiplash and Wicked. The model is able to correctly identify which musical instruments are being played in the video.
> >
> > iii) **providing an example with an unedited, natural video from youtube**. We provide an example from a random youtube video (third example in the section "Temporal Question Answering on natural videos”) on the demo website that captures cross-modal and temporal understanding. OMCAT is able to successfully “predict the activity happening in the video before the man starts speaking” and “temporally identify what is happening in the video in the second half.”
> >
> > >**Additionally, are there any known limitations in OCTAV’s data quality that might impact OMCAT’s generalizability?**
> >
> > One limitation of the dataset is the absence of overlapping audio tracks in the dataset, such as a person speaking while birds screech in the background—situations that often happen in real-life settings. Moreover, the videos in the OCTAV dataset are currently less than 1 minute long, which limits the model's ability to understand transitions in long videos. In the future, we aim to improve the dataset and refine the OMCAT model to better handle these challenges.
> >
> > **References:**
> >
> > [1] Hackl, Veronika, Alexandra Elena Müller, Michael Granitzer, and Maximilian Sailer. "Is GPT-4 a reliable rater? Evaluating consistency in GPT-4's text ratings." In Frontiers in Education, vol. 8, p. 1272229. Frontiers Media SA, 2023.
> >
> > [2] Majumder, Navonil, Chia-Yu Hung, Deepanway Ghosal, Wei-Ning Hsu, Rada Mihalcea, and Soujanya Poria. "Tango 2: Aligning diffusion-based text-to-audio generations through direct preference optimization." In Proceedings of the 32nd ACM International Conference on Multimedia, pp. 564-572. 2024.
> >
> > [3]Lee, Harrison, Samrat Phatale, Hassan Mansoor, Kellie Ren Lu, Thomas Mesnard, Johan Ferret, Colton Bishop, Ethan Hall, Victor Carbune, and Abhinav Rastogi. "Rlaif: Scaling reinforcement learning from human feedback with ai feedback." (2023).
> >
> > [4]Lipping, Samuel, Parthasaarathy Sudarsanam, Konstantinos Drossos, and Tuomas Virtanen. "Clotho-aqa: A crowdsourced dataset for audio question answering." In 2022 30th European Signal Processing Conference (EUSIPCO), pp. 1140-1144. IEEE, 2022.
> >
> > [5]Chu, Yunfei, Jin Xu, Xiaohuan Zhou, Qian Yang, Shiliang Zhang, Zhijie Yan, Chang Zhou, and Jingren Zhou. "Qwen-audio: Advancing universal audio understanding via unified large-scale audio-language models." arXiv preprint arXiv:2311.07919 (2023).

---

> > > ### Author Response · Authors · 2024-11-24
> > > **Follow-up**
> > >
> > > Dear Reviewer Q1EU,
> > >
> > > Thank you again for your feedback! As the end of the discussion period is near, kindly let us know if there are any other comments that you would like us to address.

---

> > > > ### Comment · Reviewer_Q1EU · 2024-11-26
> > > >
> > > > Thank you for your hard work. While most of my concerns are addressed, based on the overall quality of the work, I am inclined to maintain my initial score unchanged.

---

### Official Review · Reviewer_wvHV · 2024-11-03

**Soundness:** 3
**Presentation:** 3
**Contribution:** 3
**Rating:** 6
**Confidence:** 4

**Summary:**

This article presents a new dataset and model aimed at addressing the issues of fine-grained audiovisual alignment and cross-modal temporal alignment. The dataset, OCTAV, is capable of capturing event transitions between audio and video. The model, OMCAT, enhances temporal grounding and computational efficiency through RoTE and RoPE, allowing OMCAT to perform well in cross-modal temporal understanding.

**Strengths:**

1. Clear writing with a logical structure.
2. Innovative ideas that incorporate the latest research findings or technologies, presenting a unique perspective.
3. Reproducible results by providing detailed experimental methods or steps, allowing other researchers to replicate the experiments.

**Weaknesses:**

1. The experimental results are insufficient, and there is no comparison regarding the understanding ability of audio.
2. How can the quality of the generated dataset be determined?
3. In the appendix, the video understanding experiments, such as MSRVTT-QA, MSVD-QA, and ActivityNet-QA, show average performance.
4. The comparative experiments in Table 8 are not entirely fair. The data used for OMCAT with only LP, WC, and V is 430.7k less than the total dataset, so the performance of the first row being worse than the second row does not necessarily prove that the improvement is due to the characteristics of the 430.7k data rather than the difference in data volume.
5. The dataset consists of artificially added audio clips that are not aligned with the visuals and are unrelated to the overall video. However, there are no annotations regarding audiovisual misalignment in the dataset. Wouldn't this affect the model's understanding ability?
6. There are no ablation studies conducted to evaluate the effects of the proposed dataset.

**Questions:**

See weakness.

---

> ### Author Response · Authors · 2024-11-21
> **Response to Reviewer wvHV**
>
> We thank the reviewer for their positive feedback and comments that will greatly help in improving this work. Below, we address all the concerns raised.
>
> >**The experimental results are insufficient, and there is no comparison regarding the understanding ability of audio.**
>
> Thank you for pointing this out. We evaluated OMCAT on the audio question-answering task using the Clotho-AQA dataset [1], where it achieved an accuracy of 57.4%, closely matching the 57.9% achieved by Qwen-7B Audio [2], a model specifically designed for audio only understanding tasks. This demonstrates that OMCAT excels not only in cross-modal and temporal tasks but also sustains strong performance in modality-specific audio understanding. We will add these results in the paper.
>
> >**How can the quality of the generated dataset be determined?**
>
> Thanks for raising this. For assessing the quality of our generated dataset, we trained OMCAT without the OCTAV dataset and compared the results of this method on the final model trained with OCTAV. As shown in the results below, training with the OCTAV dataset leads to a gain in performance not only on the OCTAV benchmark but also across temporal understanding tasks (e.g. Charades-STA).
>
> | Model                        | Charades-STA  | OCTAV-ST-Youcook2 |
> |-------------------------|------------------ |------------------------- |
> | OMCAT w/o OCTAV  |           30.8        |                 9.9              |
> | OMCAT w/ OCTAV    |           32.3        |                 16.9            |
>
> Moreover, the QA task in OCTAV is closely tied to the ground-truth video and audio time-stamped annotations, both of which are human annotated, providing a solid foundation for quality control.
>
> Furthermore, the GPT-4-assisted generation heavily relies on prompt tuning and in-context examples (see Appendix B.1 and B.2). This approach effectively minimizes the risk of errors or hallucinations in question-answer generation, delivering reliable quality without the need for extensive human validation.
>
> >**In the appendix, the video understanding experiments, such as MSRVTT-QA, MSVD-QA, and ActivityNet-QA, show average performance.**
>
> Our work primarily aims to advance model performance in cross-modal and temporal understanding tasks. Many multimodal LLMs, such as Video LLaMA 2, leverage vast amounts of video-text paired data (around 12 million pairs), which boosts their video understanding capabilities. In contrast, OMCAT is trained on a much smaller subset of video-text data (about 2 million pairs), which naturally limits its video understanding performance. Despite this, OMCAT demonstrates competitive results in video understanding tasks and excels at fine-grained cross-modal understanding, demonstrating strong adaptability even with significantly less data.
>
> >**The comparative experiments in Table 8 are not entirely fair. The data used for OMCAT with only LP, WC, and V is 430.7k less than the total dataset, so the performance of the first row being worse than the second row does not necessarily prove that the improvement is due to the characteristics of the 430.7k data rather than the difference in data volume.**
>
> We conducted preliminary experiments to evaluate the impact of audio-video-text paired data on alignment tuning. Initially, we tried adding an additional 300k video-text pairs, but this resulted in no significant improvement in model performance. In contrast, incorporating audio-video-text paired data proved highly beneficial, significantly enhancing performance on cross-modal tasks. We will clarify this in the paper.
>
> >**The dataset consists of artificially added audio clips that are not aligned with the visuals and are unrelated to the overall video. However, there are no annotations regarding audiovisual misalignment in the dataset. Wouldn't this affect the model's understanding ability?**
>
> Thanks for raising this. The ability of the model is not affected, as we demonstrate its effectiveness on natural videos from the UnAV-100 dataset in Table 6, last row of the main paper.
>
> Additionally, the challenge is to teach the network anchoring and thus we do not believe that artificial audio-visual misalignment causes an issue as also illustrated by the demo which we perform on a youtube video updated on the demo website https://om-cat.github.io/.
>
> We updated the section “Temporal Question Answering on natural videos” in the demo website to show the performance of OMCAT on out-of-distribution videos from youtube. We provide an example from a random youtube video (third example in the section “Temporal Question Answering on natural videos”) that captures cross-modal and temporal understanding. OMCAT is able to successfully “predict the activity happening in the video before the man starts speaking” and “temporally identify what is happening in the video in the second half.”

---

> > ### Author Response · Authors · 2024-11-21
> > **Response to Reviewer wvHV**
> >
> > >**There are no ablation studies conducted to evaluate the effects of the proposed dataset.**
> >
> > Thank you for pointing this out.
> >
> > One of the ablation studies we conducted focuses on evaluating the impact of dataset size on model performance. Specifically, we use only 50% of the QA pairs from the OCTAV-ST dataset, the version of the dataset that contains single-turn question-answer pairs. This reduced dataset resulted in a 14.6% performance accuracy on the OCTAV-ST Youcook2 benchmark, compared to the 16.9% accuracy achieved when trained on the full dataset, as shown in Table 5.
> >
> > Moreover, we trained OMCAT without the OCTAV dataset and compared the results of this method on the final model trained with OCTAV. As shown in the results below, training with the OCTAV dataset leads to a gain in performance not only on the OCTAV benchmark but also across temporal understanding tasks (e.g. Charades-STA).
> > | Model                        | Charades-STA  | OCTAV-ST-Youcook2 |
> > |-------------------------|------------------ |------------------------- |
> > | OMCAT w/o OCTAV  |           30.8        |                 9.9              |
> > | OMCAT w/ OCTAV    |           32.3        |                 16.9            |
> >
> > **References**
> >
> > [1]Lipping, Samuel, Parthasaarathy Sudarsanam, Konstantinos Drossos, and Tuomas Virtanen. "Clotho-aqa: A crowdsourced dataset for audio question answering." In 2022 30th European Signal Processing Conference (EUSIPCO), pp. 1140-1144. IEEE, 2022.
> >
> > [2]Chu, Yunfei, Jin Xu, Xiaohuan Zhou, Qian Yang, Shiliang Zhang, Zhijie Yan, Chang Zhou, and Jingren Zhou. "Qwen-audio: Advancing universal audio understanding via unified large-scale audio-language models." arXiv preprint arXiv:2311.07919 (2023).

---

> > > ### Author Response · Authors · 2024-11-24
> > > **Follow-up**
> > >
> > > Dear Reviewer wvHV,
> > >
> > > Thank you again for your feedback! As the end of the discussion period is near, kindly let us know if there are any other comments that you would like us to address.

---

> > ### Comment · Reviewer_wvHV · 2024-11-27
> >
> > Thank you for addressing my concerns, most of which have been resolved. So I will increase my score.
> >
> > However, I have another question. For example, the current dataset is created by adding a random sound event after removing the original audio between the timestamps. Why didn't you consider splicing the corresponding video along with the audio from the same source, but only injecting the audio before? Would there be any improvement if the spliced video and audio are kept consistent? Although I have decided to increase the score, I hope the authors can discuss this issue. This will not affect my score. If there is not enough time left for experiments, could you find some related literature to support your views?

---

> > > ### Author Response · Authors · 2024-11-28
> > > **Response to feedback**
> > >
> > > Thank you for your feedback and for improving the score—we truly appreciate it!
> > >
> > > This is an excellent question. The primary challenge in using audio directly from the video lies in the lack of ground-truth annotations for audio events in the datasets we currently utilize. These annotations are essential for temporally anchoring the events. One possible approach is to leverage pre-trained audio models to generate audio captions for these videos. However, this method introduces considerable noise into the generated dataset, an issue we intend to address in future work.
> > >
> > > Additionally, incorporating audio events from the same video could serve as an augmentation to the OCTAV dataset, potentially enhancing cross-modal and fine-grained understanding tasks. As shown in [1], using large amounts of joint audio-video instruction-tuning data from different sources leads to significant gains in performance, however they lack in the cross-modal reasoning as we propose. We plan to explore this further in our future efforts.
> > >
> > > [1]Cheng, Zesen, Sicong Leng, Hang Zhang, Yifei Xin, Xin Li, Guanzheng Chen, Yongxin Zhu et al. "VideoLLaMA 2: Advancing Spatial-Temporal Modeling and Audio Understanding in Video-LLMs." arXiv preprint arXiv:2406.07476 (2024).

---

> > > > ### Comment · Reviewer_wvHV · 2024-11-29
> > > >
> > > > Thank you for the author's response, I would like to discuss a few points:
> > > > 1. in my understanding, only injecting audio can improve the temporal comprehension of the audio modality. So, does the dataset overlook the part of video temporal understanding? Is it possible that video also applies to this dataset design pattern?
> > > > 2. Regarding the data scale, what are the author's thoughts on the current dataset scale and future scale expansion?
> > > > 3. As a large-scale dataset that can contribute to the community, could you add what additional scenarios these datasets are suitable for? Are there any more potential applicable research fields of OCTAV in the future?

---

> > > > > ### Author Response · Authors · 2024-11-30
> > > > > **Response to Comment**
> > > > >
> > > > > > in my understanding, only injecting audio can improve the temporal comprehension of the audio modality. So, does the dataset overlook the part of video temporal understanding? Is it possible that video also applies to this dataset design pattern?
> > > > >
> > > > > Not at all, the events in the video are also temporally annotated, hence it also supports temporal understanding in the video modality (for instance, we can ask the query "what is happening in the video between o and 15 seconds?"). Our dataset thus enforces three main aspects: 1) temporal understanding of video events, 2) temporal understanding of audio and, 3) cross-modal anchored understanding of audio and video simultaneously.
> > > > >
> > > > > > Regarding the data scale, what are the author's thoughts on the current dataset scale and future scale expansion?
> > > > >
> > > > > We believe that the current scale of the dataset is quite significant when compared to other instruction-tuned datasets in the literature, including diverse video and audio domains. However, increasing both dataset scale and model scale will lead to performance improvements of OMCAT. In the future, we aim to expand the dataset in size, domain and complexity.
> > > > >
> > > > > > As a large-scale dataset that can contribute to the community, could you add what additional scenarios these datasets are suitable for? Are there any more potential applicable research fields of OCTAV in the future?
> > > > >
> > > > > Thanks for raising this! Besides fine-grained temporal and cross-modal understanding, the multi-modal temporal nature of the OCTAV dataset has wide applications in generative tasks such as textual-controlled video generation, audio generation and video to audio generation. We will look at the applicability of OCTAV datasets to such tasks in the future.

---

### Official Review · Reviewer_dpC8 · 2024-11-03

**Soundness:** 3
**Presentation:** 2
**Contribution:** 2
**Rating:** 5
**Confidence:** 4

**Summary:**

The paper proposes a new method for audio-visual question-answering. Concretely, it proposes a new method for generating a synthetic audio-vision instruction-tuning dataset, OCTAV. The method, OMCAT, is a multi-modal large language model employing staged training. The innovation is a new way of interleaving tokens and time embedding. The experiments compare the proposed method with a few recent MLLMs.

**Strengths:**

- The authors' construction of OCTAV to capture the transition between audio and video is interesting and seems effective. A lot of methods in literature create videos that have time-aligned audio and video data that correspond to the same event. Here, the authors insert arbitrary audio into the video and pose new problems for training.

**Weaknesses:**

- The method innovation is somehow mediocre. The RoTE over RoPE seems to be a small but effective improvement. However, given the recent MLLM progress, the staged training approach seems standard. Interleaving tokens is effective, but others are also employing this technique.

- The evaluation doesn’t fully support the claims that OMCAT has achieved significant gains in temporal reasoning and cross-modal alignment. First, a lot of entries in Table 5 are missing. This is quite unfortunate to draw a well-rounded comparison. Second, among them, AVSD and zero-shot Music-AVQA have relatively more methods benchmarked. On AVSD, a few methods outperform OMCAT by a 3-4% margin. On zero-shot Music-AVQA, OMCAT outperforms the second-best by a 1.6% margin.

- Some claims were exaggerated. For example, “OCTAV dataset addresses all the above mentioned limitations and provides a comprehensive benchmark for interwoven and fine-grained audio-visual understanding.” It is not a very convincing comparison to other datasets since OCTAV doesn’t collect new videos or new annotations. Later, the method training still needs to be performed on a range of audio-video datasets besides OCTAV. The combined training also shows that L84-85 in the Introduction is exaggerated, saying, "Despite this artificial setup, our experiments show that a model trained with this data performs well in naturally occurring video and audio pairs."

**Questions:**

Please see comments in the Weaknesses section. Also:
- L155-156, there are typos and much confusion. What is the unit for m? What is the rationale behind "ensuring that the videos are not too far apart and their length is not too long?"

---

> ### Author Response · Authors · 2024-11-21
> **Response to Reviewer dpC8**
>
> We thank the reviewer for their positive feedback and comments that will greatly help in improving this work. Below, we address all the concerns raised.
>
> >**The method innovation is somehow mediocre. The RoTE over RoPE seems to be a small but effective improvement. However, given the recent MLLM progress, the staged training approach seems standard. Interleaving tokens is effective, but others are also employing this technique.**
>
> We agree that interleaving tokens is effective but due to its computational complexity, we address these challenges with the effective improvement of RoTE. RoTE is model-agnostic, easy to implement, and optimized for efficient training without increasing computational demands. We highlight its importance through multiple experiments and demonstrate its impact in improving performance (see Table 5, 6 and 7).
>
> Furthermore, we agree that combining diverse training data—image-text, audio-text, video-text, and audio-video-text —is a standard approach, and we do not claim it as our unique contribution. However, we emphasize that it is a critical part of our training strategy to ensure exceptional results across a range of cross-modal tasks. We respectfully disagree with any notion that the method is limited or mediocre. Benchmark results clearly demonstrate OMCAT’s adaptability and effectiveness, significantly advancing fine-grained multi-modal understanding in the audio-video domain.
>
> >**The evaluation doesn’t fully support the claims that OMCAT has achieved significant gains in temporal reasoning and cross-modal alignment. First, a lot of entries in Table 5 are missing. This is quite unfortunate to draw a well-rounded comparison. Second, among them, AVSD and zero-shot Music-AVQA have relatively more methods benchmarked. On AVSD, a few methods outperform OMCAT by a 3-4% margin. On zero-shot Music-AVQA, OMCAT outperforms the second-best by a 1.6% margin.**
>
> Most of the models in Table 5 are either not open sourced (AVLLM, AVicuna) or do not provide pre-trained models for evaluation (PandaGPT, MacawLLM). Consequently, we trained the most recent AV model, GroundingGPT, from scratch for a fine-tuned and zero-shot evaluation and tested VideoLLaMA 2 using their pre-trained model as their full training code is not open-sourced.
>
> As both GroundingGPT and VideoLLaMA 2 are state-of-the-art models in audio-video understanding, we can safely set them as strong baselines for a direct comparison.
>
> For other benchmarks (AVSD, Music-AVQA, AVQA) which are quite widely used, we report results from other methods directly on these tasks.
>
> >**Some claims were exaggerated. For example, “OCTAV dataset addresses all the above mentioned limitations and provides a comprehensive benchmark for interwoven and fine-grained audio-visual understanding.” It is not a very convincing comparison to other datasets since OCTAV doesn’t collect new videos or new annotations.**
>
> The video datasets available in literature consist of videos from various domains and realistic scenarios, making it redundant to collect new videos. However, a primary limitation in the existing datasets listed in Table 3 is the absence of timestamped, cross-modal annotations as proposed in our OCTAV dataset.  Hence, our main contribution lies in defining a new task to enhance future research.
>
> Our pipelined approach addresses this gap by carefully selecting videos with clear event transitions and supplementing them with corresponding audio events, ensuring comprehensive cross-modal alignment in the most efficient way. We will clarify this in the main paper.

---

> > ### Author Response · Authors · 2024-11-21
> > **Response to Reviewer dpC8**
> >
> > >**Later, the method training still needs to be performed on a range of audio-video datasets besides OCTAV. The combined training also shows that L84-85 in the Introduction is exaggerated, saying, "Despite this artificial setup, our experiments show that a model trained with this data performs well in naturally occurring video and audio pairs."**
> >
> > We want to clarify that training on a range of datasets is essential for adapting the model to a wide range of audio-video tasks. However, for the task of cross-modal and temporal understanding, the model trained on artificially interleaved audio with video in the OCTAV dataset, still achieves strong performance on “natural” videos in the UnAV-100 dataset (Table 6, last row), which features real-time audio-visual events.
> >
> > Moreover, we updated the section “Temporal Question Answering on natural videos” in the demo website: https://om-cat.github.io/ to show the performance of OMCAT on out-of-distribution videos from youtube. Specifically, the first two examples show the capability of OMCAT model to identify musical capabilities on movie trailers from Whiplash and Wicked. The model is able to correctly identify which musical instruments are being played in the video.
> >
> > Furthermore, we also provide an example from a random youtube video (third example in the section “Temporal Question Answering on natural videos”) that captures cross-modal and temporal understanding. OMCAT is able to successfully “predict the activity happening in the video before the man starts speaking” and “temporally identify what is happening in the video in the second half.”
> >
> > >**L155-156, there are typos and much confusion. What is the unit for m? What is the rationale behind "ensuring that the videos are not too far apart and their length is not too long?"**
> >
> > The unit for m is in seconds. Specifically, m makes sure that a single event in the video is no longer than 10 seconds (a hyperparameter chosen to filter very long events) and T limits the overall length of the video sample with multiple event transitions to 30 seconds. We will clarify this in the main paper.

---

> > > ### Author Response · Authors · 2024-11-24
> > > **Follow-up**
> > >
> > > Dear Reviewer dpC8,
> > >
> > > Thank you again for your feedback! As the end of the discussion period is near, kindly let us know if there are any other comments that you would like us to address.

---

### Official Review · Reviewer_XvaG · 2024-11-04

**Soundness:** 3
**Presentation:** 4
**Contribution:** 3
**Rating:** 8
**Confidence:** 3

**Summary:**

This paper targets solving fine-grained, cross-modal temporal understanding, particularly when correlating events across audio and video streams. To better tackle this task, they introduce the Omni Context Aware Transformer(OMCAT), together with a dataset Omni Context and Temporal Audio Video(OCTAV). OMCAT integrates time embedding and is proposed to improve temporal localization and computational efficiency through RoTE (Rotary Time Embeddings).OCTAV is designed to capture the transformation of audio and video events, helping the model to better understand the temporal order of events and cross-modal associations. Comprehensive experiments and ablation studies are provided to demonstrate that OMCAT significantly improves temporal reasoning and cross-modal alignment.

**Strengths:**

The paper addresses a clear gap in cross-modal temporal understanding, providing new opportunities for research in integrating audio and video modalities and giving related solutions and datasets.

The approach is soundness:
1. The use of RoTE (Rotary Time Embeddings) as an extension of RoPE to enhance temporal grounding is a creative advancement that could inspire further research in temporal embeddings.
2. The multi-training-stage ensures thorough model development and evaluation

The experiment comparison is comprehensive. experiments and ablation studies provide strong evidence for the model’s state-of-the-art performance on Audio-Visual Question Answering (AVQA) tasks.

**Weaknesses:**

1. The synthetic nature of the OCTAV dataset may limit the performance of the model in real-world applications.

2. The technical details of methods such as RoTE are not explained in detail, which may affect understanding and reproducibility.

3. RoTE is claimed to have a better ability to capture actual elapsed between frames while lacking more comparison of how fine-grand it can do.

4. The dataset is comprehensive but including also the evaluation dataset. This raises the concern of proposed methods.

**Questions:**

You mentioned the ability to process music. Could you show some result of out-of-distribution results in the event-intensive data like movie trailers?

Audio tracks often overlap. Have you considered the relevant design?

Video and music or video often have strong coupling or delay. Have you considered how to find the time node of multi-modal events? For example, due to broadcasting issues, the time node of the ball landing and the time node of the sound in a tennis video may not be synchronized. So I ask which mode should be prioritized when the tennis ball lands?

---

> ### Author Response · Authors · 2024-11-21
> **Response to Reviewer XvaG**
>
> We thank the reviewer for their positive and insightful feedback and comments that will greatly help in improving this work. Below, we address all the concerns raised.
>
> >**The synthetic nature of the OCTAV dataset may limit the performance of the model in real-world applications.**
>
> Real-world applications can be a bit ambiguous, and hence we address this concern by:
>
> i) **providing a benchmark on "natural" videos**. We evaluated the model on the UnAV-100 dataset (refer to Table 6, last row) in the main paper, which features real-time audio-visual events. Although the annotation style in UnAV-100 differs somewhat from our proposed OCTAV dataset, it still offers a robust benchmark for testing cross-modal and temporal understanding. As demonstrated in the example provided in the appendix (Figure 7), UnAV-100 dataset has complex, dynamic, real-world scenarios, making it a good testbed for real-world applications.
>
> ii) **providing an example with a movie trailer**. We updated the section “Temporal Question Answering on natural videos” in the demo website: https://om-cat.github.io/ to capture this. Specifically, the first two examples show the capability of OMCAT model to identify musical capabilities on movie trailers from Whiplash and Wicked. The model is able to correctly identify which musical instruments are being played in the video.
>
> iii) **providing an example with an unedited, natural video from youtube**. We provide an example from a random youtube video (third example in the section "Temporal Question Answering on natural videos”) on the demo website that captures cross-modal and temporal understanding. OMCAT is able to successfully “predict the activity happening in the video before the man starts speaking” and “temporally identify what is happening in the video in the second half.”
>
> We can hence show OMCAT's suitability for real-world applications.
>
> >**The technical details of methods such as RoTE are not explained in detail, which may affect understanding and reproducibility.**
>
> We have added an algorithm section in Appendix B for better understanding of RoTE. Moreover, we will make our implementation publicly available for reproducibility.
>
> >**RoTE is claimed to have a better ability to capture actual time elapsed between frames while lacking more comparison of how fine-grained it can do.**
>
> RoTE’s capacity to capture fine-grained information depends on two things:
>
> i) **Sampling intervals for video and audio frames.** In our setup, we sample 64 uniformly spaced video frames and 3-second audio windows, enabling RoTE to process fine details effectively within videos up to one minute long. For longer videos, this approach becomes less granular—a limitation we plan to address in future work and,
>
> ii) **Training data.** As for the dataset, there is a lack of an existing fine-grained dataset that fully captures the nuances of real-world scenarios. This gap presents an opportunity for future work, where developing such a dataset could provide an even more robust benchmark for evaluating models like OMCAT.
>
> >**The dataset is comprehensive but including also the evaluation dataset. This raises the concern of proposed methods.**
>
> The evaluation split of OCTAV dataset is the same as the eval split of Youcook2 and ActivityNet. Additionally, we would like to point out that for a fair comparison, we trained and evaluated the baseline models on our proposed versions of the dataset similar to OMCAT to assess the ability of OMCAT. As demonstrated in Tables 5 and 6, OMCAT significantly outperforms these baselines on the OCTAV benchmark.

---

> > ### Comment · Reviewer_XvaG · 2024-11-26
> > **Feed Back to Author**
> >
> > Dear Author,
> >
> > Thanks for your feedback. From my perspective, the answer remains unclear in the potential application, which leads to the novelty and necessity of this general task. Further more, left the experiment design of this workroom for improvement. In deed, since there are so many MLLM papers each year, I prefer to see how this work can help the community, or can this task be helpful in specific areas. For example, why do we need to detect the music events, what is its inner connection with visual context, and how are so many modalities combined with each other?
> >
> > However, this paper is well structured, and the experiments are comprehensive. So I decided to keep my rating.
> >
> > Best
> > XvaG

---

> > > ### Author Response · Authors · 2024-11-26
> > > **Response to Feedback**
> > >
> > > Thank you for the rating and providing feedback on our responses, we greatly appreciate it.
> > > Compared to all the existing MLLMs, what sets OMCAT apart is it's ability to:
> > > 1. **temporally identify audio and visual events** which is extremely useful for generating high quality synthetic data with fine-grained temporal control for video/audio generation. All the other MLLMs generate global short captions for a video and are not sensitive to the time the event is happening.
> > > 2. **anchor a user's query on a specific modality**, which offers a unique cross-modal experience (e.g. for users that can only listen and not see and vice versa).
> > > 3. **music understanding**. Labelling music videos with high quality captions is expensive making it a bottleneck for music generation. OMCAT can address this by generating music related captions that can then be used for generating high-quality music. However, in a special case e.g., a complex musical performance scene which involves multiple instruments (sounding and non-sounding), it is difficult to analyze the sound by using the audio modality alone, hence using both auditory and visual modalities is extremely effective [1].
> > >
> > > References:
> > >
> > > [1] Liu, Xiulong, Zhikang Dong, and Peng Zhang. "Tackling data bias in music-avqa: Crafting a balanced dataset for unbiased question-answering." In Proceedings of the IEEE/CVF Winter Conference on Applications of Computer Vision, pp. 4478-4487. 2024.

---

> ### Author Response · Authors · 2024-11-21
> **Response to Reviewer XvaG**
>
> >**You mentioned the ability to process music. Could you show some result of out-of-distribution results in the event-intensive data like movie trailers?**
>
> We updated the section “Temporal Question Answering on natural videos” in the demo website: https://om-cat.github.io/ to clarify this. Specifically, the first two examples show the capability of OMCAT model to identify musical capabilities on movie trailers from Whiplash and Wicked. The model is able to correctly identify which musical instruments are being played in the video.
>
> Furthermore, we also provide an example from a random youtube video (third example in the section “Temporal Question Answering on natural videos”) that captures cross-modal and temporal understanding. OMCAT is able to successfully “predict the activity happening in the video before the man starts speaking” and “temporally identify what is happening in the video in the second half.”
>
> >**Audio tracks often overlap. Have you considered the relevant design?**
>
> Overlapping audio tracks (e.g., a man speaking with birds screeching in the background) represents a very realistic scenario and is an important consideration for future work. We plan to extend the OCTAV dataset and the OMCAT model to address this problem in the future.
>
> >**Video and music or video often have strong coupling or delay. Have you considered how to find the time node of multi-modal events? For example, due to broadcasting issues, the time node of the ball landing and the time node of the sound in a tennis video may not be synchronized. So I ask which mode should be prioritized when the tennis ball lands?**
>
> In cases where there is a timing mismatch between audio and visual signals, it’s essential to consider the specific task at hand. For tasks where accurate synchronization is crucial, such as in video-to-audio generation models, this lag in the training data can be detrimental. In our setup, we can address this by enabling the model to respond to queries such as, “when can we see the tennis ball landing?” and “when can we hear the tennis ball landing?” This dual-query approach allows us to pinpoint and evaluate any misalignment between the two modalities and act accordingly.

---

> > ### Author Response · Authors · 2024-11-24
> > **Follow-up**
> >
> > Dear Reviewer XvaG,
> >
> > Thank you again for your feedback! As the end of the discussion period is near, kindly let us know if there are any other comments that you would like us to address.

---

### Meta-Review · Area_Chair_NKL1 · 2024-12-21

**Metareview:**

The paper presents a dataset OCTAV and a model OMCAT for solving the audio-visual question answering task. The dataset OCTAV is generated from time-stamped videos by artificially inserting transition audio between clips, followed by generating QA pairs using a GPT model -- the inserted audio is claimed to allow the model to capture the temporal order of events in the video. OMCAT is essentially a multimodal transformer but uses rotary position embeddings (RoTE) that captures absolute time using a parameterization inspired by a mechanical clock. Experiments are provided on AVQA, Music-AVQA, and AVSD dataset as well as the Charades-STA, YouCook2, and ActivityNet benchmarks and demonstrate promising results.

**Additional Comments On Reviewer Discussion:**

The paper received four reviews, with two reviewers suggesting accept or borderline accept and two below borderline. The submission received a significant engagement between the authors and the reviewers during the discussion phase. Reviewers liked the organization of the paper and the quality of the writing, while agreeing that the paper attempts to address a clear gap in state-of-the-art cross-modal temporal understanding. However, there are important concerns raised, namely:
1. limited innovation in the methodology (dpC8)
2. missing state-of-the-art evaluations (dpC8, wvHV, XvaG),  exaggerated claims (dpC8), and potential for data contamination (XvaG)
3. The use of GPT for data generation and no quality control of its responses (Q1EU).

For 1., the reviewers point out that introduction of RoTE over RoPE, which is the key novelty in the paper, is incremental. Further, authors agree during the discussion that RoTE can handle upto 1 minute videos, however longer videos need further adjustments.

For 2., authors provided some clarification on missing evaluation mainly due to the unavailability of the state-of-the-art models, while confirmed that they carefully address the data splits used in the training set of OMCAT avoiding data contamination.

For 3., the authors described the use of GPT as a standard practice. Although no human validation is done, which could be expensive, however authors indirectly validate this by claiming that training with OCTAV improves the performance of their model against training without OCTAV, showing that GPT provided annotations are useful.

Even after the discussion phase, the reviewers stand split. Despite the method's impressive improvements in performance reported across a wide array of datasets, AC concords with the issues pointed out by the reviewers that: 1) there appears to be a lack of any solid technical contribution the paper makes that is responsible for the solid results, 2) there are missing comparisons as pointed out by dpC8, wvHV, XvaG and that brings a lack of clarity to where the method stands against prior approaches, and 3) the OCTAV dataset is produced by putting together several other datasets with heuristic modifications that lack any specific insights and synthetic annotations produced without any quality control. AC also notes that the most positive reviewer (XvaG) shows reservations in the recommendation. As such, AC recommends reject.

---

### Decision · Program_Chairs · 2025-01-22

Reject